# Targeting IRE1α improves insulin sensitivity and thermogenesis and suppresses metabolically active adipose tissue macrophages in male obese mice

Dan Wu[1,2], Venkateswararao Eeda[2], Zahra Maria[2], Komal Rawal[2], Audrey Wang[3], Oana Herlea-Pana[2], Ram Babu Undi[4], Hui-Ying Lim[1,4], Weidong Wang[1,2]*

[1]Department of Genetics, Heersink School of Medicine, UAB Comprehensive Diabetes Center, University of Alabama at Birmingham, Birmingham, United States; [2]Department of Medicine, Division of Endocrinology, The University of Oklahoma Health Sciences Center, Oklahoma City, United States; [3]Indian Springs School, Pelham, United States; [4]Department of Physiology, Harold Hamm Diabetes Center, The University of Oklahoma Health Sciences Center, Oklahoma City, United States

*For correspondence: ww8@uab.edu

## eLife Assessment

The study presents **important** findings on inositol-requiring enzyme (IRE1α) inhibition on diet-induced obesity (overnutrition) and insulin resistance where IRE1α inhibition enhances thermogenesis and reduces the metabolically active and M1-like macrophages in adipose tissue. The evidence supporting the conclusions is **convincing**. The work will be of interest to cell biologists and biochemists working in metabolism, insulin resistance and inflammation with a broad eLife readership.

**Abstract** Overnutrition engenders the expansion of adipose tissue and the accumulation of immune cells, in particular, macrophages, in the adipose tissue, leading to chronic low-grade inflammation and insulin resistance. In obesity, several proinflammatory subpopulations of adipose tissue macrophages (ATMs) identified hitherto include the conventional 'M1-like' CD11C-expressing ATM and the newly discovered metabolically activated CD9-expressing ATM; however, the relationship among ATM subpopulations is unclear. The ER stress sensor inositol-requiring enzyme 1α (IRE1α) is activated in the adipocytes and immune cells under obesity. It is unknown whether targeting IRE1α is capable of reversing insulin resistance and obesity and modulating the metabolically activated ATMs. We report that pharmacological inhibition of IRE1α RNase significantly ameliorates insulin resistance and glucose intolerance in male mice with diet-induced obesity. IRE1α inhibition also increases thermogenesis and energy expenditure, and hence protects against high fat diet-induced obesity. Our study shows that the 'M1-like' CD11c+ ATMs are largely overlapping with but yet non-identical to CD9+ ATMs in obese white adipose tissue. Notably, IRE1α inhibition diminishes the accumulation of obesity-induced metabolically activated ATMs and 'M1-like' ATMs, resulting in the curtailment of adipose inflammation and ensuing reactivation of thermogenesis, without augmentation of the alternatively activated M2 macrophage population. Our findings suggest the potential of targeting IRE1α for the therapeutic treatment of insulin resistance and obesity.

## Introduction

Overnutrition and obesity are closely associated with insulin resistance which serves as a common precursor to type 2 diabetes, cardiovascular diseases, non-alcoholic fatty liver disease, and other metabolic disorders. Under chronic energy overload, adipose tissue expands significantly for the storage of the excess energy in the form of lipids and this expansion coincides with the recruitment and accumulation of immune cells, in particular macrophages, in adipose tissue. Once the lipid storage capacity is reached, adipocytes can no longer incorporate lipids from the circulation but instead release fatty acids into the circulation through lipolysis. Meanwhile, the excess accumulation of immune cells in the adipose tissue leads to chronic low-grade inflammation, including aberrant proinflammatory cytokine production, in the adipose tissue (*Weisberg et al., 2003*; *Xu et al., 2003*). Both aberrant adipocyte lipolysis and adipose tissue inflammation are known to exacerbate each other (*Foley et al., 2021*) and are regarded as key factors contributing to insulin resistance in obesity.

The accumulated adipose tissue macrophages (ATMs) were initially thought to undergo a phenotypic polarization from an anti-inflammatory M2 to a pro-inflammatory M1 state, promoting local and systemic inflammation with the production and secretion of pro-inflammatory cytokines (*Lumeng et al., 2007*). However, there is a growing body of evidence that does not support this obesity-induced M1/M2 polarization model (*Xu et al., 2013*; *Kratz et al., 2014*; *Coats et al., 2017*; *Hill et al., 2018*; *Jaitin et al., 2019*). Instead, recent studies have identified a predominant population of ATMs that emerge in obesity and present a metabolically activated phenotype (*Xu et al., 2013*; *Kratz et al., 2014*; *Coats et al., 2017*). The metabolically activated ATMs have lately been shown to express CD9 and/or Trem2 by studies using unbiased single-cell RNA sequencing (*Hill et al., 2018*; *Jaitin et al., 2019*). These obesity-associated metabolically active ATMs not only potentiate inflammation but also promote dead adipocyte clearance (*Kratz et al., 2014*; *Coats et al., 2017*; *Hill et al., 2018*). However, the relationship between the previously identified 'M1-like' CD11c+ATMs and the new CD9+ and/or Trem2+ ATM subpopulations is unclear.

The endoplasmic reticulum (ER) is a major site for protein synthesis and folding and the synthesis of lipid and sterol. When the demand for protein folding in the ER exceeds its capacity, ER stress occurs. The three ER stress sensors, inositol-requiring enzyme (IRE)–1, PKR-like ER-regulating kinase (PERK), and activating transcription factor (ATF)–6, are then activated to trigger the unfolded protein response (UPR) that initially serves as an adaptive means to resolve ER stress, but will eventually become maladaptive if the perturbation persists. IRE1α, the most evolutionarily conserved among the UPR sensors, is an ER transmembrane protein with dual serine/threonine kinase and RNase domains and once activated, can catalyze the unconventionally cleavage of *Xbp1* mRNA for XBP1 translation or initiate a process termed regulated IRE1-dependent decay of RNA (RIDD; *Fonseca et al., 2011*; *Back and Kaufman, 2012*; *Papa, 2012*; *Hetz et al., 2013*). IRE1α activation was reported to induce/potentiate inflammatory cytokine production and inflammation through the control of transcriptional regulation and intracellular signaling pathways via its dual enzymatic activities (*Martinon et al., 2010*; *Lerner et al., 2012*; *Oslowski et al., 2012*). ER stress and IRE1α activation has been reported in both human and rodent obese adipose tissues (*Boden et al., 2008*, *Gregor et al., 2009*) and in the stromal vascular fraction (SVF) of adipose tissue where ATMs are enriched (*Shan et al., 2017*). Recent studies have implicated ER stress as an early event of nutrient excess in the development of insulin resistance (*Ozcan et al., 2004*; *Boden et al., 2008*, *Cnop et al., 2012*; *Kawasaki et al., 2012*). Accordingly, targeting ER stress/IRE1α has been proposed as a direction for the improvement of insulin resistance in obesity and diabetes (*Hetz et al., 2013*; *Marciniak et al., 2022*).

However, thus far, the exact role of IRE1α on the adipose function and remodeling remains inconclusive. For example, adipocyte-specific knockout of IRE1α was shown to promote brown adipose tissue (BAT) activation and browning of white adipose tissue (WAT) and alleviate diet-induced obesity and insulin resistance by suppressing its RNase-mediated degradation of the mRNA of thermogenic gene *Ppargc1a* and hence elevating UCP1 level (*Chen et al., 2022*). On the other hand, other studies reported that the IRE1α-XBP1 pathway is required for the transcriptional induction of UCP1 in brown adipocyte in vitro (*Asada et al., 2015*) and is indispensable for proper adipogenesis (*Sha et al., 2009*). Additionally, inhibiting IRE1α kinase activity was found to be sufficient enough to block the inflammation-induced lipolysis in adipocytes (*Foley et al., 2021*), which contrasts with the observation that adipocyte deletion of IRE1α promotes β3-AR agonist-stimulated WAT lipolysis as reported (*Chen et al., 2022*).

The function of IRE1α on adipose tissue remodeling and insulin resistance has also been shown through its activity in ATMs. Deletion of IRE1α in myeloid lineage including macrophage was recently shown to prevent mice from diet-induced obesity and insulin resistance (*Shan et al., 2017*). The IRE1α deletion was reported to achieve this by reversing the obesity-induced M1/M2 polarization (*Shan et al., 2017*), leading to the augmented M2 population in adipose tissue that induces BAT activation and WAT remodeling through the synthesis and action of catecholamines (*Nguyen et al., 2011*; *Shan et al., 2017*). However, it was recently reported that the alternatively activated M2 macrophages do not synthesize catecholamines or contribute to adipose tissue adaptive thermogenesis (*Fischer et al., 2017*). Moreover, as mentioned above, the obesity-associated ATMs were recently shown to not undergo the classical M1/M2 switch but to adopt a metabolic activation state (*Xu et al., 2013*; *Kratz et al., 2014*; *Hill et al., 2018*; *Jaitin et al., 2019*). It is therefore unclear what changes in the ATM under IRE1α deletion are accountable for the protection against obesity and insulin resistance as observed in the myeloid-specific IRE1α knockout model (*Shan et al., 2017*). It is unknown whether IRE1α inhibition has any effect on the obesity-induced metabolically activated ATM population.

In this study, we used a pharmacological inhibitor of IRE1α STF-083010 (STF) to investigate the effect of IRE1α inhibition on insulin resistance and obesity in the adult male mice. We observed that STF treatment improves insulin sensitivity and protects against weight gain in diet-induced obesity (DIO) mice. IRE1α inhibition also reverses the high fat diet (HFD)-induced whitening of BAT and WAT remodeling and increases thermogenesis. Notably, STF treatment suppressed the M2 ATM subpopulation in the DIO mice. Our studies also showed that IRE1α inhibition diminishes the obesity-elicited pro-inflammatory ATM subpopulations including the newly identified CD9+ ATMs and adipose inflammation which is known to suppress thermogenesis (*Sakamoto et al., 2013*; *Goto et al., 2016*; *Sakamoto et al., 2016*). In all, our study provides new insights into the mechanisms by which IRE1α inhibition protects against obesity and insulin resistance and serves as a foundation for targeting IRE1α as a therapeutic means in the treatment of obesity and insulin resistance.

## Results

### IRE1α RNase inhibition ameliorates insulin resistance in mice with diet-induced obesity

We first examined whether the ER stress response/UPR pathway was altered in the adipose tissues of animals with obesity. In C57BL/6 J mice fed high fat diet for 14 weeks, we observed that the mRNA levels of genes representing all three branches of UPR [spliced *Xbp1* (*Xbp1s*) for IRE1α branch, *Atf4* and *Ddit3* for PERK branch, and *Edem1* and *Hspa5* for ATF6 branch] were up-regulated in both epididymal white adipose tissues (eWAT) (*Figure 1A*) and inguinal white adipose tissue (iWAT) (*Figure 1—figure supplement 1A*). As IRE1α activation is associated with the IRE1-dependent decay of mRNA (RIDD) activity in which IRE1 cleaves mRNAs (*Hollien and Weissman, 2006*), we analyzed the mRNA levels of two RIDD targets *Blos1* and *Col6a1* by qRT-PCR. We observed that *Blos1* and *Col6a1* mRNA levels were decreased in the eWAT (*Figure 1B*) while *Col6a1* mRNA level was also down-regulated in the iWAT (*Figure 1—figure supplement 1B*) of HFD-fed mice. These results suggest the over-nutrition/obesity triggers ER stress and IRE1α activity in adipose tissue.

We then asked whether IRE1α activation contributes to obesity-induced insulin resistance and importantly whether pharmacological inhibition of IRE1α can reverse the pre-existing obesity-induced insulin resistance. We used STF-083010 (STF), a member of hydroxy-aryl-aldehyde class of IRE1α RNase inhibitors that engage the RNase-active site of IRE1a with high affinity and specificity by exploiting a shallow complementary pocket through pi-stacking interactions with His910 and Phe889 and an essential Schiff base interaction between the aldehyde moiety of the inhibitor and the side chain amino group of Lys907 (*Sanches et al., 2014*), to treat diet-induced obesity (DIO) mice with established insulin resistance. Male C57BL/6 J mice fed on HFD for 8 weeks gained significant glucose intolerance (*Figure 1—figure supplement 1C*) and were treated with STF at 10 mg/kg of BW once daily for 4 weeks (*Figure 1—figure supplement 1D*) via IP injection, a dose previously shown to significantly inhibits IRE1α RNase activity in vivo (*Papandreou et al., 2011*; *Tufanli et al., 2017*; *Herlea-Pana et al., 2021*), while still on HFD. We observed that STF treatment improved glucose tolerance with lower peak glucose level and decreased AOC (area of the curve) in response to a bolus of exogenous glucose compared to their vehicle-treated counterpart (*Figure 1C–C'*). STF treatment significantly

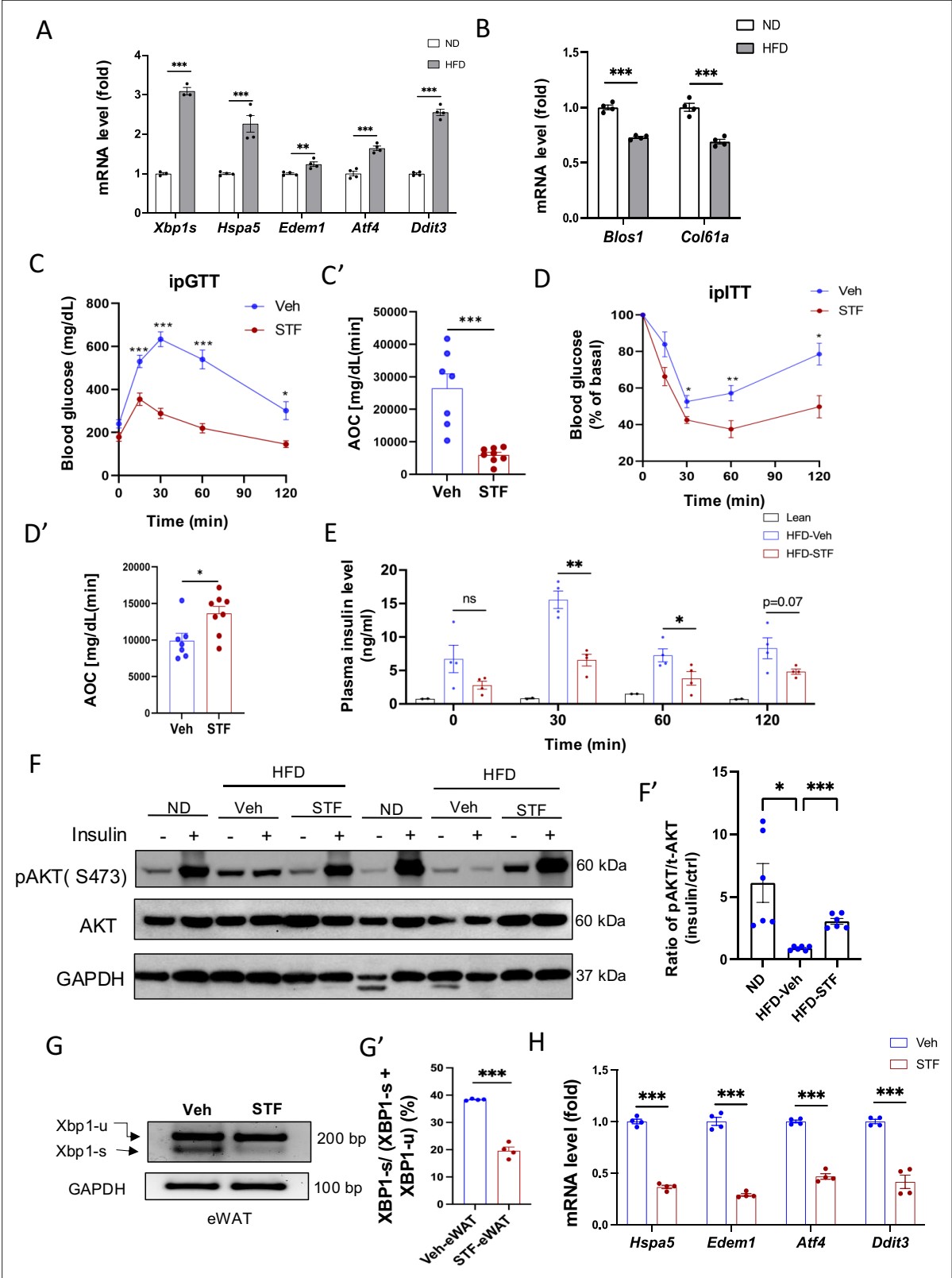

**Figure 1.** IRE1α inhibitor STF improves glucose tolerance and insulin sensitivity in DIO mice. (**A**) mRNA levels of ER stress genes in the eWAT from mice fed ND (normal diet) and HFD for 14 weeks, assessed by qRT-PCR. The results are expressed as fold change and are representative of 3 independent experiments. (**B**) mRNA levels of IRE1α RIDD target genes in the eWAT from mice fed ND and HFD for 14 weeks, assessed by qRT-PCR. The results are expressed as fold change and are representative of 3 independent experiments. (**C-C′**) Glucose tolerance test on mice treated with vehicle (n=7)

*Figure 1 continued on next page*

*Figure 1 continued*

or STF (n=8) at two weeks of treatment. Blood glucose levels (**C**) measured at indicated time points after intraperitoneal injection of glucose (1.5 g/kg body weight) following 6 h fasting and AOC area of the curve, (**C'**). (**D–D'**) Insulin tolerance test on mice treated with vehicle (n=7) or STF (n=8) at three weeks of treatment. Blood glucose levels (**C**) normalized to basal level at indicated time points after intraperitoneal injection of insulin (1.2 IU/kg body weight) following 6 h fasting and the AOC (area of the curve, D'). (**E**) Plasma insulin levels at indicated time points after intraperitoneal injection of glucose (3 g/kg body weight) following 6 h fasting at 3.5 weeks of treatment. **F-F'**. Protein levels of pAKT (Ser473) and total AKT in eWAT of ND mice or DIO mice treated with vehicle or STF, analyzed with Western blotting (**F**). GAPDH as a loading control. Fasted mice were IP injected with either saline or insulin and scarified after 15 min and eWATs were collected. The data shown are representative of 3 independent experiments. Quantitative analysis of immunoblots was performed (**F'**). **G-G'**. Splicing of *Xbp1* mRNA was analyzed from eWAT of DIO mice treated with vehicle or STF by RT-PCR and resolved by agarose gel electrophoresis. The full length (unspliced, XBP1-u) and spliced (XBP1-s) forms of *Xbp1* mRNA were indicated (**G**) and quantified with t-XBP1 as the sum of XBP1-u and XBP1-s (**G'**). GAPDH mRNA was used as an internal control. The data shown are representative of 3 independent experiments. (**H**) mRNA levels for indicated genes were analyzed in eWAT of DIO mice treated with vehicle or STF by qRT-PCR. The results were expressed as fold change and were representative of 3 independent experiments. Data were expressed as mean ± SEM and analyzed using the unpaired two-tailed Student's t-test between two samples or ANOVA with multiple comparisons. ∗$P<0.05$, ∗∗$P<0.01$, and ∗∗∗$P<0.001$.

The online version of this article includes the following source data and figure supplement(s) for figure 1:

**Source data 1.** Western blot membrane images and Xbp1 splicing gel image in eWAT without labeling.

**Source data 2.** Western blot membrane images and Xbp1 splicing gel image in eWAT with labeling.

**Figure supplement 1.** ER stress and IRE1α activation in adipose tissues in DIO mice.

**Figure supplement 1—source data 1.** Xbp1 splicing gel image in iWAT without labeling.

**Figure supplement 1—source data 2.** Xbp1 splicing gel image in iWAT with labeling.

improved insulin sensitivity in DIO mice (*Figure 1D–D'*). We also observed markedly decreased resting and glucose-stimulated plasma levels of insulin in DIO mice treated with STF (*Figure 1E*), indicative of increased insulin sensitivity in the STF-treated mice. To investigate whether the increased insulin sensitivity by STF is reflected in insulin signaling, we examined the phosphorylation status of Akt, an effector of insulin signaling pathway, in the adipose tissue. Contrary to the significant enhancement in the insulin-stimulated Akt phosphorylation at Ser473 observed in the epididymal white adipose tissue (eWAT) of non-HFD lean animals, the insulin-stimulated Akt phosphorylation was blunted in the eWAT of DIO mice. However, STF treatment significantly reversed the insulin-stimulated Akt phosphorylation (*Figure 1F–F'*). Collectively, these data indicate that STF improves insulin sensitivity and glucose control under HFD.

To investigate whether the STF amelioration of insulin resistance and obesity reflects on its inhibition of IRE1α activity in adipose tissue, we examined the status of IRE1α-mediated *Xbp1* mRNA splicing in WAT from STF-treated DIO mice and observed that as assessed by RT-PCR followed by electrophoretic separation, the level of spliced *Xbp1* (*Xbp1s*) mRNA was significantly reduced in both eWAT (*Figure 1G–G'*) and iWAT (*Figure 1—figure supplement 1E-E"*) from STF-treated animals relative to vehicle group. Moreover, the mRNA levels of *Xbp1* target genes *Hspa5* and *Edem1* were highly suppressed in both eWAT and iWAT from STF-treated mice (*Figure 1H*, *Figure 1—figure supplement 1F*). Interestingly, we also observed the decrease in the mRNA levels of *Atf4* and *Ddit3*, key components of the PERK pathway in eWAT and iWAT of STF-treated DIO mice (*Figure 1H*, *Figure 1—figure supplement 1F*). The effect of STF treatment on PERK pathway in the adipose tissue probably reflects the positive feedback loops among the branches of UPR under in vivo conditions as previously reported (*Tsuru et al., 2013*; *Márton et al., 2017*; *Kapuy et al., 2020*).

## STF protects against diet-induced obesity and obesity-associate morbidities

We observed that while vehicle-treated mice continued to gain weight on HFD, 4 week STF treatment slightly decreased the body weight (*Figure 2A* and *Figure 4—figure supplement 1A*), with no apparent food intake difference between the vehicle and STF groups (*Figure 4—figure supplement 1B*). Body composition analysis using EchoMRI also revealed that STF treatment resulted in a significant decrease in the fat mass but without significant changes in the lean mass (*Figure 2B and C*) in the HFD-fed mice. We next analyzed the size of adipocytes in WAT sections and observed that adipocytes were significantly smaller from STF-treated DIO mice than that of vehicle-treated ones (*Figure 2C and F*) as shown in both average diameters (*Figure 2D,G*) and frequencies in large adipocytes (*Figure 2E and H*) in both eWAT and inguinal WAT (iWAT). The average area of adipocytes was

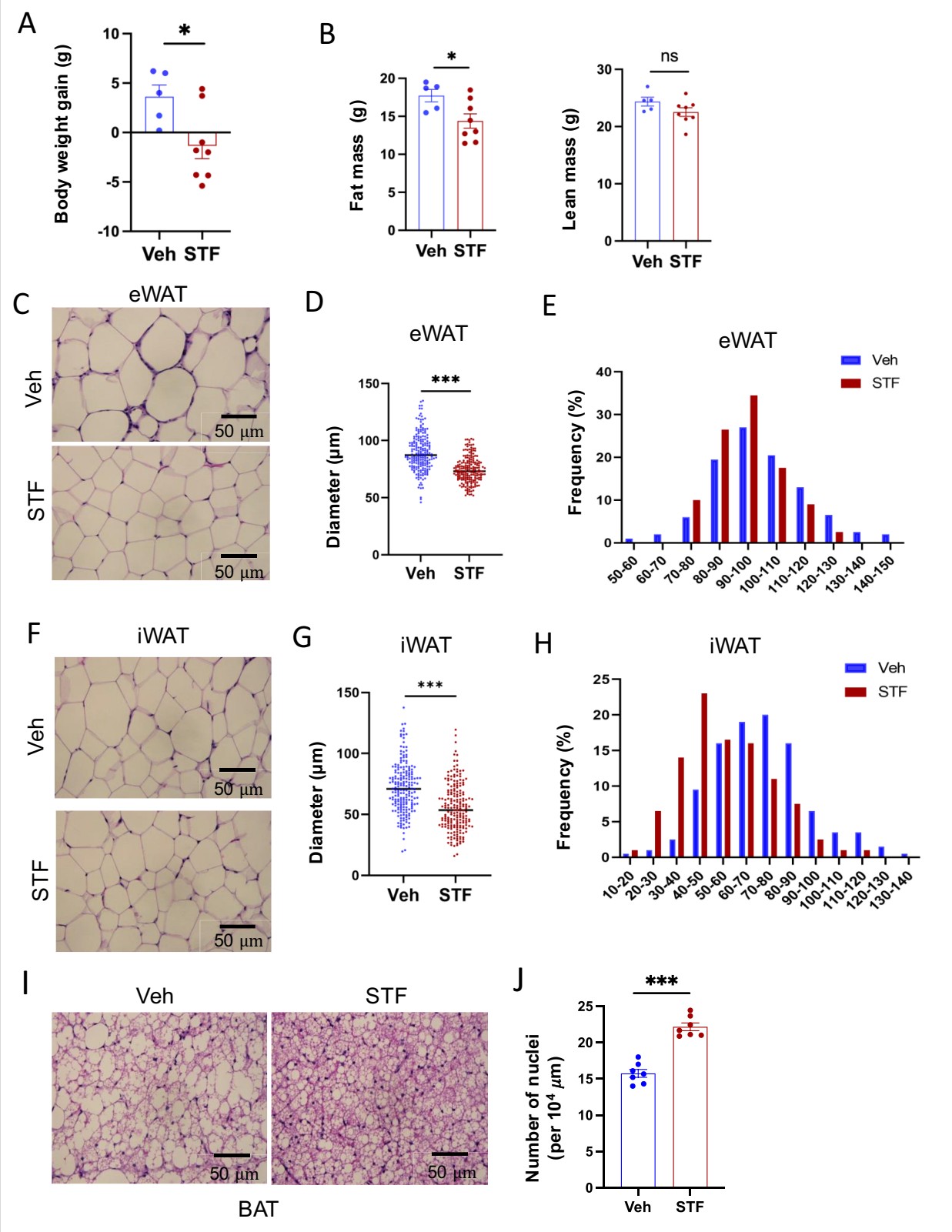

**Figure 2.** STF protects against diet-induced adiposity and obesity. (**A**) Body weight gain of DIO mice treated with vehicle (n=5) or STF (n=8) for 4 weeks. (**B**) Fat mass and lean mass of DIO mice treated with vehicle or STF, assessed at the end of 4-week treatment by EchoMRI. (**C–E**) eWAT adipocytes. Representative images of H&E staining of eWAT sections (**C**). Average diameter of adipocytes (µM)/field (**D**) and size distribution (**E**) in eWAT. (**F–H**) iWAT adipocytes. Representative images of H&E staining of iWAT sections (**F**). Average diameter of adipocytes (µM)/field (**G**) and size distribution (**H**) in iWAT.

*Figure 2 continued on next page*

*Figure 2 continued*

(I) Representative images of H&E staining of BAT. (J) Measurement of number of BAT nuclei. Data were expressed as mean ± SEM and analyzed using the unpaired two-tailed Student's t-test between two samples or ANOVA with multiple comparisons. *p<0.05, **p<0.01, and ***p<0.001.

similarly significantly shrunk in the eWAT (*Figure 4—figure supplement 1D*) and iWAT (*Figure 4— figure supplement 1E*) of DIO mice treated with STF relative to vehicle treatment. In addition, there was a significant reduction in lipid droplets in the BAT from STF-treated DIO mice (*Figure 2I–J*).

A common comorbidity of obesity and insulin resistance is metabolic dysfunction-associated steatotic liver disease (MASLD; *Khan et al., 2019*). We therefore investigated the potential effects of STF on liver steatosis in obesity. First, we examined the liver histology in haematoxylin and eosin- (H.E.) stained sections. We observed the presence of numerous lipid droplets in the liver, a hallmark of live steatosis, in the vehicle-treated DIO mice compared to that in the BL/6 mice fed normal chow (*Figure 3A*). Strikingly, STF treatment markedly reduced the number of the lipid droplets in the liver of DIO mice compared to vehicle treatment (*Figure 3A and A'*). We next examined the expression of genes involved in lipogenesis in the liver of the STF- and vehicle-treated DIO mice and found that the lipogenic genes FASN, stearoly-coA desaturase-1 (*Scd1*), and acetyl coA-carboxylase (*Acc*) were down-regulated in the liver of STF-treated DIO mice compared to vehicle-treated mice (*Figure 3B*). Furthermore, because MASLD is associated with an impaired suppression of hepatic glucose output (*Marchesini et al., 2001*), we assessed the expression of genes involved in gluconeogenesis, including *glucose-6-phosphatase catalytic subunit* (*G6pc1*) and *phosphoenolpyruvate carboxykinase* (*Pepck*). We found that the expression level of *G6pc1* in the liver from STF-treated DIO mice was also significantly decreased compared to that of vehicle-treated ones while there was no effect on *Pepck* expression (*Figure 3C*). MASLD is also characterized by excessive inflammation, in particular macrophage accumulation, in addition to lipid deposition (*Lefere and Tacke, 2019*). We therefore investigated the effect of STF on macrophage population in the obese liver and observed that STF treatment suppressed the macrophage accumulation of DIO mice, as marked by the F4/80 immunofluorescent staining (*Figure 3D–E*). Furthermore, obesity is tightly associated with dyslipidemia. We therefore examined the effect of STF on dyslipidemia and observed that STF treatment decreased the plasma levels of triglyceride, free fatty acid (FFA), and total cholesterol (*Figure 3F–H*).

## STF treatment promotes thermogenesis and energy expenditure

The observation that STF treatment protects against HFD-induced obesity (*Figure 2A*) suggests an increase in energy expenditure (EE) in these mice. We therefore investigated this in DIO mice treated with STF. We observed that average oxygen consumption ($VO_2$) was higher in DIO mice treated with STF than with vehicle, using indirect calorimetry (*Figure 4A–B* and *Figure 4—figure supplement 1F-F'*). Similarly, the average carbon dioxide production ($VCO_2$) was significantly increased in STF-treated DIO mice (*Figure 4C–D* and *Figure 4—figure supplement 1G-G'*). As a result, we found that STF treatment moderately but significantly increased mean (EE) compared to vehicle treatment (*Figure 4E–F* and *Figure 4—figure supplement 1H-H'*), without alterations in respiratory exchange ratio (RER) value or physical activities (*Figure 4G–H*).

To ask whether the energy expenditure increase by STF is attributable to thermogenic heat production, we investigated the effect of STF treatment on the activation status of brown adipose tissue (BAT) in DIO animals. BAT is an organ specialized for energy expenditure through the action of mitochondrial uncoupling protein 1 (UCP1). During obesity, lipid-droplet content is increased in BAT (whitening of BAT), which impairs its ability to dissipate the energy. We observed that STF treatment reversed the whitening of BAT, as reflected in much less lipid droplet intensity in BAT, in DIO mice, compared to vehicle treatment (*Figure 2I*). Immunofluorescent staining showed the elevated staining intensity of UCP1 protein in BAT per unit area in DIO mice compared to vehicle treatment (*Figure 4I–K*). Furthermore, STF-treated mice showed increased expression of genes involved in thermogenic and mitochondrial functions, including *Prdm16*, *Ucp1*, *Cidea*, *Ppara*, *Dio2*, *Cox5b*, and *Ppargc1a* (*Pgc1α*), in BAT (*Figure 4L*). These data thereby suggest that STF promotes BAT activation. Induction of beige fat in the WAT also plays important role in the adaptive thermogenesis and energy expenditure (*Kajimura et al., 2015*). We therefore examined the effect of STF on the status of beiging activation in the WAT. The immune-signal intensity of UCP1 protein was markedly increased in the eWAT (*Figure 4M* and *Figure 4—figure supplement 2A*) and iWAT (*Figure 4—figure supplement*

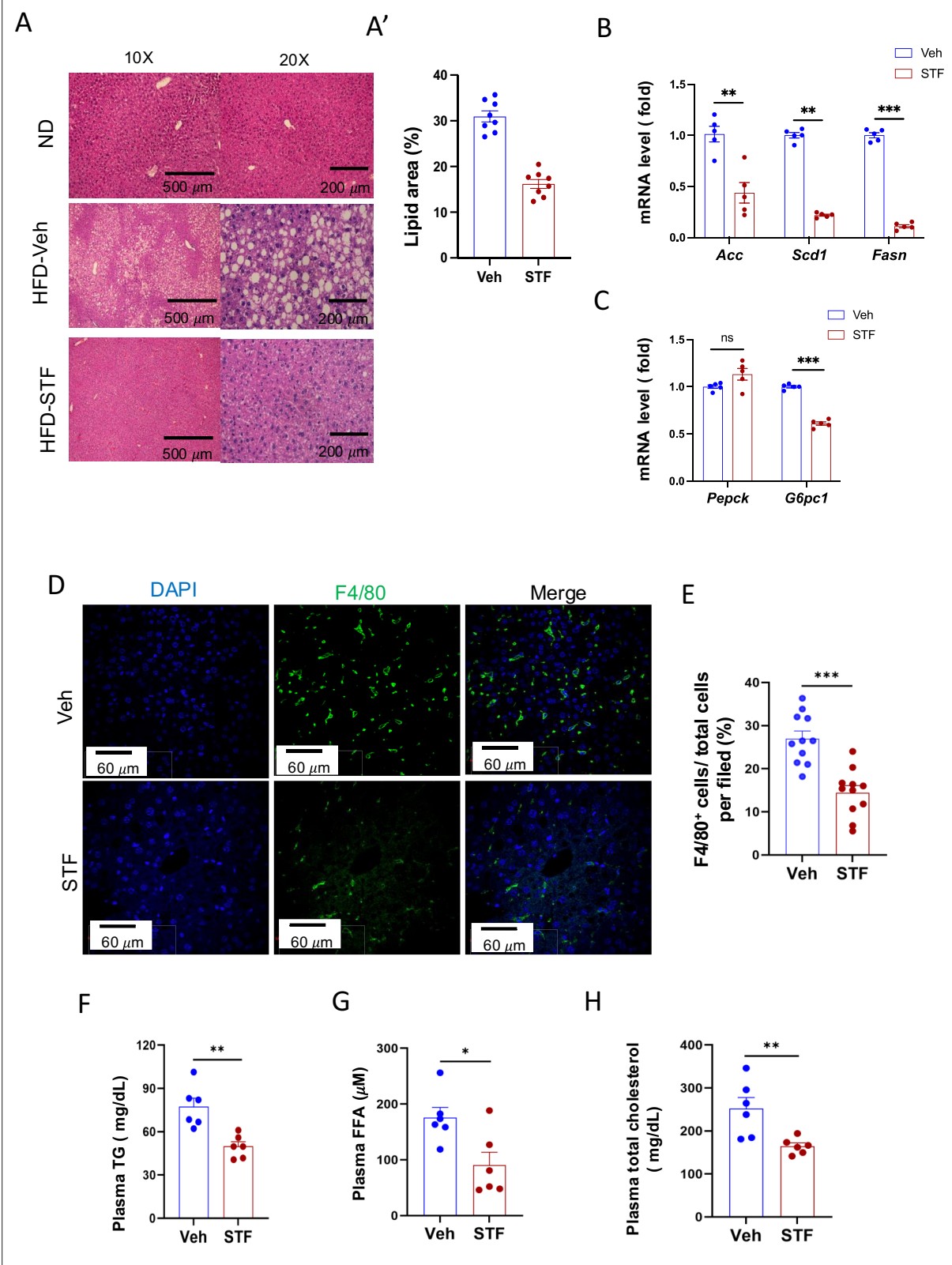

**Figure 3.** STF treatment improves liver steatosis and dyslipidemia in DIO mice. (**A-A'**) Liver histology. Representative images of H&E of liver sections from mice of ND and HFD treated with vehicle or STF (**A**). The quantification of lipid area in liver samples (**A'**). (**B**) mRNA levels of lipogenic genes in livers from DIO mice treated with vehicle or STF, assessed by qRT-PCR. The results are expressed as fold change and are representative of three independent experiments. (**C**) mRNA levels of gluconeogenic genes in livers from DIO mice treated with vehicle or STF, assessed by qRT-PCR. The

*Figure 3 continued on next page*

*Figure 3 continued*

results are expressed as fold change and are representative of three independent experiments. (**D–E**) Immunofluorescent staining of F4/80 in liver. Representative images of staining (**D**) and percentage of number of F4/80-positive cells over total cells/field (**E**). (**F–H**) The levels of serum TG (**F**), FFA (**G**) and total cholesterol (**H**) were analyzed by respective ELISA kits. Blood samples were collected upon euthanization at the end of treatment. Data were expressed as mean ± SEM and analyzed using the unpaired two-tailed Student's t-test between two samples or ANOVA with multiple comparisons. *p<0.05, **p<0.01, and ***p<0.001.

*2B-C*) of DIO mice treated with STF. In addition, the expression of thermogenic genes was significantly increased in both eWAT and iWAT of DIO mice subjected to STF treatment compared to vehicle (*Figures 4N and 3D*). These results suggest the involvement of beige fat activation in thermogenesis and energy expenditure in STF-treated mice. Moreover, we also investigated the effect of STF on the expression -adrenoceptors, which play important role in adaptive thermogenesis in response to cold stimulation, on adipose tissues and observed that STF treatment elevated the levels of both β2-adrenoceptor (*Adrb2*) and β3-adrenoceptor (*Adrb3*) in BAT (*Figure 4O*) and the levels of *Adrb3* in iWAT and eWAT (*Figure 4—figure supplement 2E-F'*).

## Effect of STF treatment on obesity-associated macrophage activation

Since adipose tissue macrophage (ATM) accumulation plays important role in contributing to insulin resistance and obesity (*Xu et al., 2003*; *Lumeng et al., 2007*), we investigate whether STF treatment impacts ATM accumulation in DIO mice as IRE1α activity was observed to be heightened in obese macrophages (*Shan et al., 2017*). During obesity, the expansion of adipose tissues accompanies the recruitment and accumulation of macrophages in the adipose tissue, and crown-like structures (CLS) are formed upon the infiltration and residence of macrophages around dead adipocytes in the adipose tissues (*Cinti et al., 2005*). As shown in *Figures 2C and 5A*, the area of CLS was drastically reduced in the eWAT of DIO mice treated with STF relative to vehicle-treated mice, although CLS area showed no significant difference in iWAT between vehicle and STF groups (*Figure 5—figure supplement 1A*). Immunofluorescent staining of F4/80, a pan-macrophage marker, was drastically increased in the eWAT under HFD, but was reversed in STF-treated mice (*Figure 5B–C*). Moreover, we examined the expression levels of pan-macrophage marker genes (*Cd68* and *F4/80 (Adgre1)*) and found that STF treatment significantly alleviated the mRNA levels of *Cd68* and *Adgre1* in the stromal vascular fraction (SVF where immune cells reside) of the eWAT (*Figure 5D*). Expectedly, STF treatment also inhibited the splicing of IRE1α target *Xbp1* mRNA in the eWAT SVF (*Figure 5—figure supplement 1B-D*). Next, we used flow cytometry to analyze the changes of ATMs by STF treatment under HFD. We observed that the percentage of macrophage population (double positive for F4/80 and CD11b markers [F4/80$^+$CD11b$^+$]) did not alter substantially in the SVF fraction isolated from eWAT under HFD compared to normal chow (*Figure 5E* and *Figure 5—figure supplement 2A*); however, given the HFD-induced expansion of the eWAT, the total number of F4/80$^+$CD11b$^+$ macrophages has increased drastically by 49.6-fold: 0.29±0.25 × 10$^5$ (normal chow, n=8) vs 1.43±0.086 × 10$^6$ (14-week HFD, n=8) cells/eWAT fat pad/mouse (*P*p0.0041) (*Table 1a*). The density of F4/80$^+$CD11b$^+$ macrophage population in eWAT also increased under HFD by 12.5-fold after normalizing the total number of macrophages to fat pad weight (0.62±0.45 × 10$^5$ (normal chow) vs. 0.77±0.015 × 10$^6$ cells (14 week HFD)/g of eWAT/mouse, p=0.0045, *Table 1a*). Excitingly, STF treatment significantly lessened the HFD-induced increases in the percentage, total number, and density of ATMs by 37.4%, 46%, and 54%, respectively (*Figure 5E* and *Figure 5—figure supplement 2A* and *Table 1a*). These results indicate that STF suppresses macrophage accumulation in the adipose tissue.

It was previously reported that macrophages that accumulated in adipose tissues undergo a 'phenotypic switch' from an anti-inflammatory M2 (alternatively activated) phenotype to a proinflammatory M1 (classically activated) state, a process believed to be critical for the development of systemic insulin resistance (*Lumeng et al., 2007*). We therefore investigated the effect of STF treatment on such an ATM switch. We observed that under HFD, the percentage (14.59% (ND) vs. 52.95% (HFD), p=0.007, *Figure 5F* and *Figure 5—figure supplement 2A*, *Table 1b*), total number/eWAT/mouse (0.25±0.16 × 10$^4$ (ND, n=8) vs 0.75±0.11 × 10$^6$ (HFD, n=8), 299.58-fold, p<0.0001, *Table 1b*), and density/g of eWAT/mouse (0.59±0.23 x 10$^4$ (ND) vs. 4.11±0.96 × 10$^5$ (HFD), 70-fold, p=0.0018, *Table 1b*) of F4/80$^+$CD11b$^+$CD11c$^+$ triple positive cells, which are typically characterized as the classical 'M1-like' adipose tissue macrophages, were markedly increased in the eWAT. Unexpectedly, we

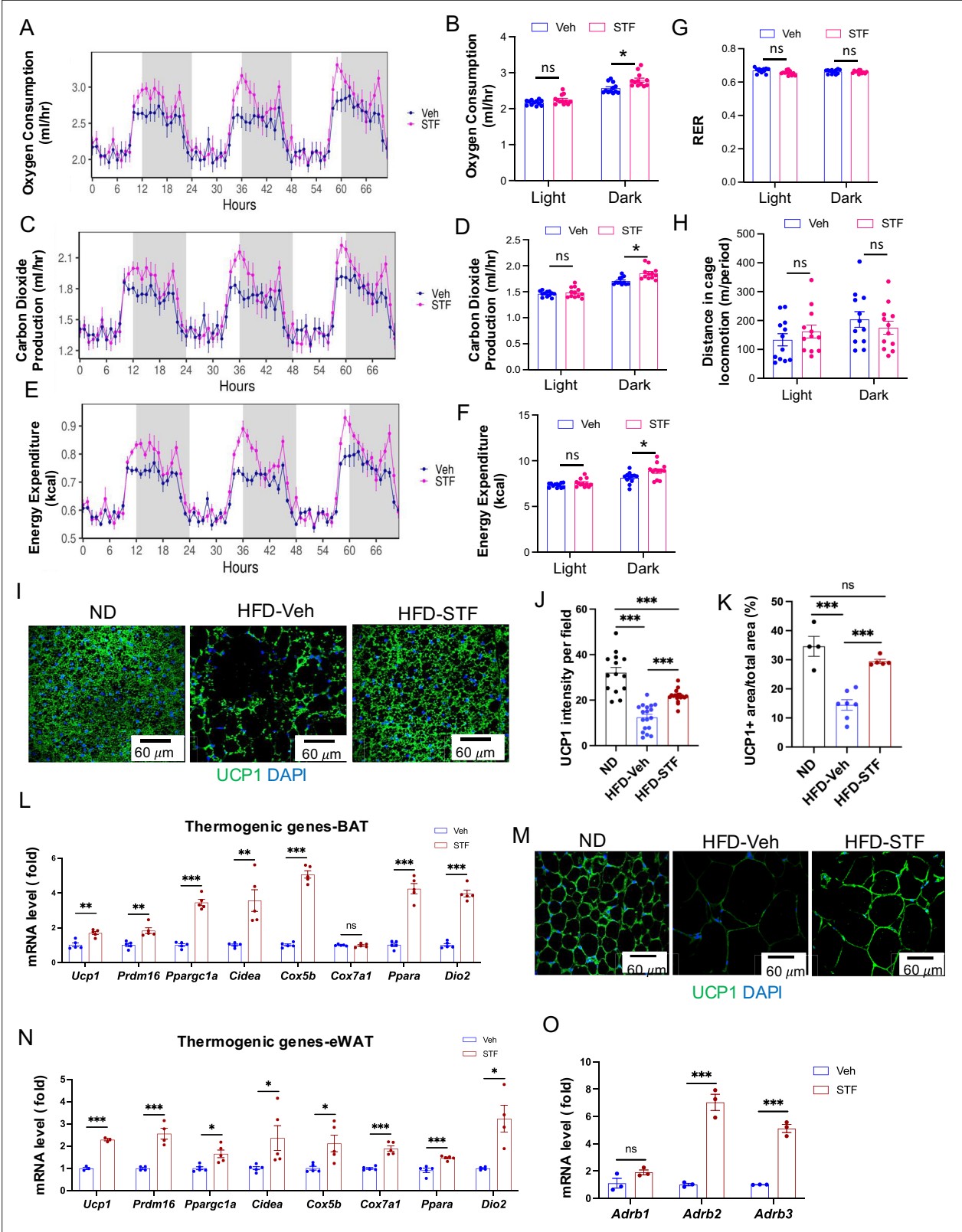

**Figure 4.** STF promotes energy expenditure and thermogenesis in DIO mice. (**A–B**) Curves (**A**) and quantification (**B**) of oxygen consumption levels. (**C-D**). Curves (**C**) and quantification (**D**) of CO2 production levels. (**E–F**) Curves (**E**) and quantification (**F**) of energy expenditure levels calculated. For A-F, the data was analyzed for group effect (ANOVA) along with the mass and interaction effects as necessary (generalized linear model, GLM). (**G**) RER (respiratory exchange ratio) level. (**H**) Daily activity (m of running distance/mouse). (**I–K**) UCP1 immunofluorescence staining of BAT sections from ND-fed

*Figure 4 continued on next page*

*Figure 4 continued*

mice and DIO mice treated with vehicle or STF. Representative images of UCP1 immunofluorescence staining of BAT (**I**), quantification of immune-signal intensity of UCP1 (**J**), and quantification of UCP1-positive area (**K**). (**L**) mRNA levels of thermogenic genes in BAT from DIO mice treated with vehicle or STF, assessed by qRT-PCR. The results are expressed as fold change and are representative of three independent experiments. (**M**) Representative images of UCP1 immunofluorescence staining of eWAT sections from ND-fed mice and DIO mice treated with vehicle or STF. (**N**) mRNA levels of thermogenic genes in eWAT from DIO mice treated with vehicle or STF, assessed by qRT-PCR. The results are expressed as fold change and are representative of three independent experiments. (**O**) mRNA levels of β-adrenoceptor genes in BAT from DIO mice treated with vehicle or STF, assessed by qRT-PCR. The results were expressed as fold change and are representative of three independent experiments. Data for G-O were expressed as mean ± SEM and analyzed using the unpaired two-tailed Student's t-test between two samples or ANOVA with multiple comparisons. $*p<0.05$, $**p<0.01$, and $***p<0.001$.

The online version of this article includes the following figure supplement(s) for figure 4:

**Figure supplement 1.** STF protects against diet-induced obesity and promotes energy expenditure.

**Figure supplement 2.** STF promotes thermogenesis in WAT of DIO mice.

found that the bulk majority of these CD11c$^+$ ATMs were also positive for CD206, a marker thought for M2 macrophages, in the eWAT under HFD (*Figure 5—figure supplement 2B*), and that approximately half the population is positive for both CD11c and CD206 in DIO ATMs, whereas there are only 8.55% CD11c$^+$CD206$^+$ ATMs under ND when gated against CD11c and CD206 (*Figure 5G* and *Figure 5—figure supplement 2A*), suggesting the enrichment of a mixed M1/M2 phenotype, which is consistent with recent reports (*Shaul et al., 2010*; *Wentworth et al., 2010*; *Sica and Mantovani, 2012*). Moreover, we found that the total number and density of CD11c$^+$CD206$^+$ ATMs increased by 337- and 75-fold respectively under HFD (*Table 1c*; ND: n=8; HFD: n=8), a greater increase than those of CD11C-postive alone (CD11c$^+$CD206$^-$) ATMs under HFD (*Supplementary file 1*; ND: n=8; HFD: n=8). On the other hand, for CD206$^+$CD11c$^-$ ATM subpopulation, a presumed M2-like ATM subpopulation which accounted for the majority of ATMs under ND as expected (*Figure 5G* and *Table 1d*), HFD drove the increase in the total cell number and density of this population by over 16- and 4-fold respectively despite the diminished percentage given the huge expansion of total ATMs (*Table 1d*; ND: n=8; HFD: n=8), although to a lesser degree than that for CD11c$^+$CD206$^+$ or CD11c$^+$CD206$^-$ macrophages (*Table 1c* and *Supplementary file 1a*). Moreover, CD206$^-$CD11c$^-$ ATM subpopulation also exhibited a trend similar to that of CD206$^+$CD11c$^-$ ATMs (*Supplementary file 1b*; ND: n=8; HFD: n=8). Our findings confirm the notion that ATMs do not follow the classic M2 to M1 switch (*Xu et al., 2013*; *Kratz et al., 2014*; *Hill et al., 2018*).

It has been reported that the germline deletion of IRE1α in myeloid lineage reverses the switch in M1/M2 polarization with the increase in M2 population to induce BAT activation and WAT remodeling and protect mice against diet-induced obesity (*Shan et al., 2017*). However, in light of increasing evidence from others (*Xu et al., 2013*; *Kratz et al., 2014*; *Hill et al., 2018*) and our current findings that do not support a HFD-triggered classical M1/M2 polarization, we sought to interrogate the effect of IRE1α inhibition on ATM subpopulations, which by far is poorly understood. In DIO mice treated with STF for 4 weeks, we found that consistent with the overall reduction of ATM population (*Figure 5E*, *Table 1a*), the percentage, total number, and density of the posited 'M1-like' F4/80$^+$CD-11b$^+$CD11c$^+$ triple positive cells were substantially diminished in the SVF of eWAT (*Figure 5F* and *Figure 5—figure supplement 2A*, *Table 1b*), but to an even greater extent than the overall ATM population. The trend in the reduction of F4/80$^+$CD11b$^+$CD11c$^+$ ATMs by STF remains similar regardless of whether CD206$^+$ was positive (F4/80$^+$CD11b$^+$CD11c$^+$CD206$^+$) or negative (F4/80$^+$CD11b$^+$C-D11c$^+$CD206$^-$) (*Figure 5G*, *Table 1c* and *Supplementary file 1a*). In addition, the total number and density of M2-like CD206$^+$CD11c$^-$ ATMs were also significantly reduced in the SVF of eWAT of STF-treated DIO animals (*Table 1d*), but to a lesser degree than CD11c$^+$ ATMs. In contrast, STF treatment even slightly increased the total number and density of the CD206 CD11c double-negative ATMs (CD206$^-$CD11c$^-$F4/80$^+$CD11b$^+$) or non-macrophage (F4/80$^-$CD11b$^-$) populations (*Supplementary file 1b and c*), although none of them reaching statistically significance.

As the 'M1-like' CD11c$^+$ and M1/M2 mixed ATMs were reported to be associated with proinflammation (*Shaul et al., 2010*; *Wentworth et al., 2010*), we examined the mRNA levels of proinflammatory cytokine genes —*Tnfa*, *Il1b*, *Il6*, and *monocyte chemoattractant protein-1* (*Ccl2*)— and found that they were down-regulated in the SVFs of eWAT from STF-treated DIO mice compared to vehicle-treated counterparts (*Figure 5H*), which correlates with our observed decreased populations

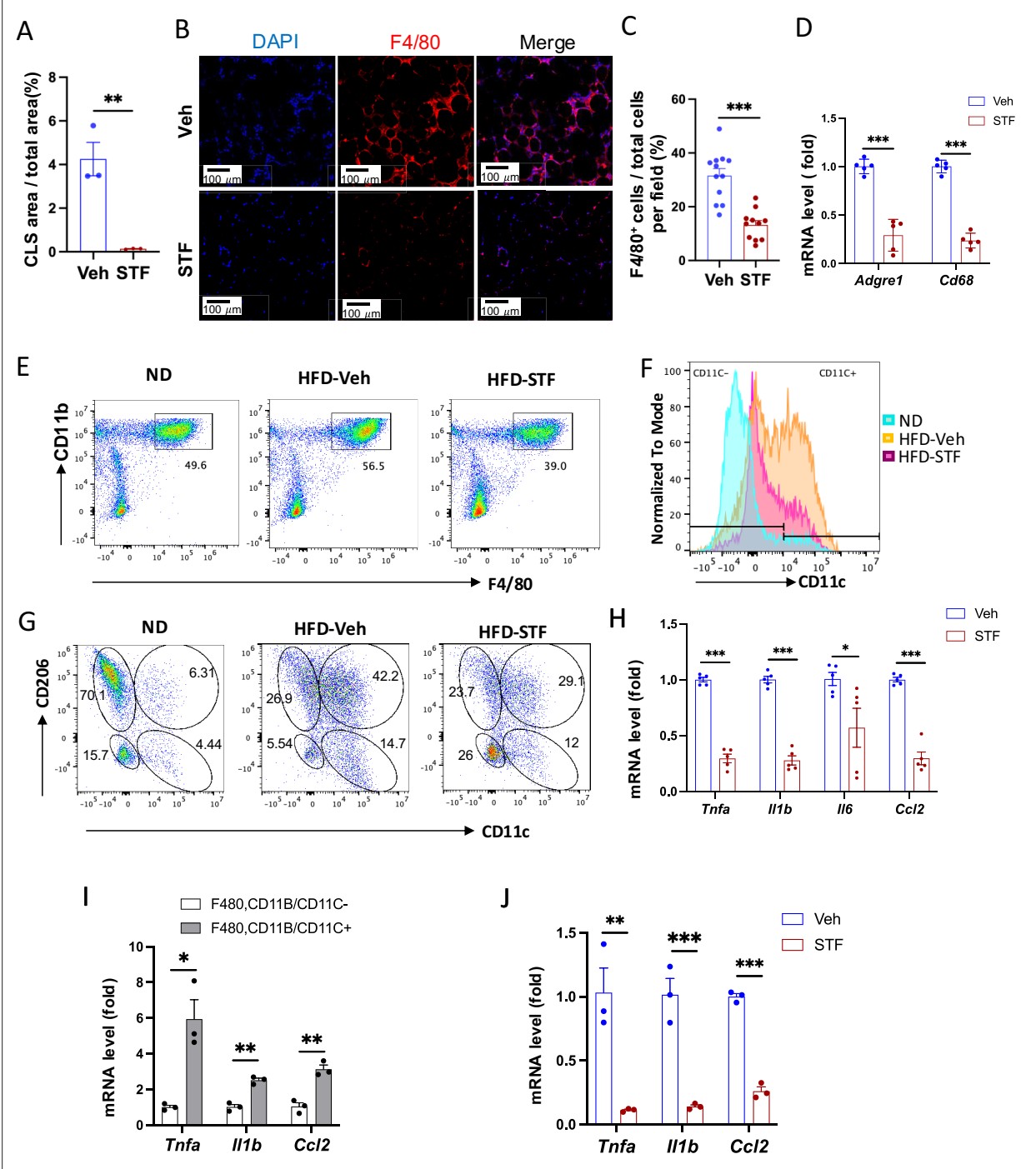

**Figure 5.** STF treatment suppresses ATM accumulation and adipose inflammation in DIO mice. (**A**) Percentage of CLS area in eWAT sections from DIO mice treated with vehicle or STF. (**B–C**) Immunofluorescent staining of F4/80 in the eWAT sections of DIO mice treated with vehicle or STF. Representative images of F4/80 in eWAT (**B**) and percentage of the number of F4/80-positive cells over total cells/field (**C**). (**D**) mRNA levels of total macrophage marker genes in the SVF of eWAT from DIO mice treated with vehicle or STF, assessed by qRT-PCR. The results are expressed as fold change and are representative of three independent experiments. (**E**) Flow cytometry of total ATMs from eWAT SVFs of ND mice (n=4) and DIO mice treated with vehicle (n=4) or STF(n=4), gated against CD11b and F4/80 antibodies. Complete gating path was shown in *Figure 5—figure supplement 2A*. (**F**) Flow cytometry analysis of CD11C-positive/negative ATMs (F4/80⁺CD11b⁺CD11C⁺/⁻) from eWAT SVFs of mice with an overlay of gated CD11C marker for ND (yellow), HFD-Veh (blue) and HFD-STF (pink) shown. Data in Y-axis was presented as 'Normalized to mode' for percentages. Complete gating path was shown in *Figure 5—figure supplement 2A*. (**G**) Flow cytometry of 'M1-like' and 'M2-like' subsets of ATMs from eWAT SVFs of ND, HFD-Veh and HFD-STF mice, gated against CD11c and CD206 antibodies. Complete gating path was shown in *Figure 5—figure supplement 2A*.

*Figure 5 continued on next page*

*Figure 5 continued*

(**H**) mRNA levels of indicated proinflammatory cytokine/chemokine genes in the SVF of eWAT of DIO mice treated with vehicle or STF, assessed by qRT-PCR. The results are expressed as fold change and are representative of three independent experiments. (**I**) mRNA levels of indicated proinflammatory cytokine/chemokine genes in the indicated sorted cells from SVFs of eWAT of DIO mice, assessed by qRT-PCR. The results are expressed as fold change and are representative of three independent experiments. (**J**) mRNA levels of indicated proinflammatory cytokine/chemokine genes in the CD11C$^+$ ATMs (F4/80$^+$CD11b$^+$CD11C$^+$) sorted from SVFs of eWAT of DIO mice treated with vehicle or STF, assessed by qRT-PCR. The results were expressed as fold change and were representative of three independent experiments. Data were expressed as mean ± SEM and analyzed using the unpaired two-tailed Student's t-test between two samples or ANOVA with multiple comparisons. *p<0.05, **p<0.01, and ***p<0.001.

The online version of this article includes the following source data and figure supplement(s) for figure 5:

**Figure supplement 1.** STF inhibits IRE1α activity in the SVF of obese eWAT.

**Figure supplement 1—source data 1.** Xbp1 splicing gel image in SVF of eWAT without labeling.

**Figure supplement 1—source data 2.** Xbp1 splicing gel image in SVF of eWAT with labeling.

**Figure supplement 2.** STF suppresses obesity-driven ATM accumulation and adipose inflammation.

of 'M1-like' CD11c$^+$ and M1/M2 mixed ATMs in the STF-treated DIO animals (*Figure 5E–F*). Similarly, STF treatment also inhibited the mRNA levels of these cytokine genes in the SVFs of iWAT (*Figure 5—figure supplement 2C*). Furthermore, in CD11c$^+$ ATMs (F4/80$^+$CD11b$^+$CD11c$^+$) sorted from the eWAT SVFs of DIO mice, which expectedly expressed higher mRNA levels of *Tnfa*, *Il1b*, and *Ccl2* than CD11c$^-$ ATMs (*Figure 5I*), STF treatment remarkably reduced the expression levels of *Tnfa*, *Il1b*, and *Ccl2* (*Figure 5J*). STF inhibited the IRE1α RNase activity in these cells as reflected by the diminished *Xbp1s* mRNA levels (*Figure 5—figure supplement 2D*). These results reveal that STF treatment suppresses the adipose tissue inflammation and the accumulation of pro-inflammatory ATM without augmenting (suppressing instead) M2-like ATMs.

## Metabolically active ATMs are suppressed in DIO mice by STF treatment

Recent studies have shown that instead of the classical proinflammatory M1 polarization, obesity drives the expansion of metabolically active heterogeneous ATM populations. These metabolically active ATMs possess heightened lysosome and lipid metabolism (*Xu et al., 2013*; *Kratz et al., 2014*; *Coats et al., 2017*), express cell surface markers CD9 or Trem2 at a higher frequency, and are pro-inflammatory (*Hill et al., 2018*; *Jaitin et al., 2019*). These studies reported that the CD9$^+$ macrophages have high amounts of intracellular lipid in lysosome-like structures, express genes related to lysosomal-dependent lipid metabolism (*Hill et al., 2018*) and represent the majority of total adipose tissue macrophages in obesity (*Jaitin et al., 2019*). Given our findings on the significant inhibitory effect of STF on ATM accumulation and inflammation, we therefore investigated whether STF impacts these alterations in the adipose tissues of DIO mice. We first determined the effect of STF treatment on the metabolic activation of ATMs. Proteins involved in lipid metabolism including ABCA1, CD36, and PLIN2 were reported to be specifically up-regulated in metabolically active macrophages in obesity (*Kratz et al., 2014*). We observed that the mRNA levels of *Abca1*, *Cd36*, and *Plin2* were elevated in the SVF portion of the eWAT in DIO mice compared to their normal diet counterparts, but STF treatment significantly suppressed the up-regulation of these genes (*Figure 6A*). Moreover, genes involved in lysosomal biogenesis and phagocytosis including *Atp6v1b2*, *Atp6v0d2*, *Lipa*, and *Lamp2* were also reported to be induced in the ATMs in obesity (*Xu et al., 2013*; *Coats et al., 2017*). We therefore examined the effect of STF on the mRNA levels of *Atp6v1b2*, *Atp6v0d2*, *Lipa*, and *Lamp2* and observed that their mRNA levels were elevated in the SVF portion of the eWAT in DIO mice and that STF treatment suppressed the up-regulation of these genes, although the suppression on Lipa does not reach statistical significance (*Figure 6B*). In addition, expression levels of other genes known to be involved in lipid metabolism and lysosomal biogenesis, including *Fabp4*, *Fabp5*, and *Stsb1*, were also suppressed in the SVF portion of the eWAT in DIO mice by the treatment of STF (*Figure 6C*). Lastly, we also observed a similar trend in the SVFs of the iWAT (*Figure 6—figure supplement 1A-B*).

We next investigated the effect of STF on the newly identified ATM subpopulations. First we observed that the mRNA levels of *CD9* and *Trem2* were elevated in the SVFs of the eWAT and iWAT in DIO mice relative to age-matched normal chow-fed mice but STF treatment suppressed the up-regulation of these genes (*Figure 6D* and *Figure 6—figure supplement 1C*). In addition,

**Table 1.** Effect of STF treatment on 'M1-like' and 'M2-like' ATMs in obesity.

(A) The total number and density of ATMs (F4/80$^+$CD11B$^+$) in eWAT in mice with ND, HFD-Veh, or HFD-STF. (B) The percentage, total cell number, and density of CD11C$^+$ ATMs in eWAT in mice with ND, HFD-Veh, or HFD-STF. (C) The total cell number and density of C11DC$^+$ CD206$^+$ATMs in eWAT in mice with ND, HFD-Veh, or HFD-STF. (D) The total cell number and density of CD206$^+$C11DC$^-$ ATMs in eWAT in mice with ND, HFD-Veh, or HFD-STF. Data in a-d were obtained from two batches of four mice each and were expressed as mean ± SEM and analyzed using the unpaired two-tailed Student's t-test between two samples or ANOVA with multiple comparisons. *p<0.05, **p<0.01, and ***p<0.001.

**A**

**F4/80+CD11B+**

|  | Total cell#/eWAT/mouse | Cell#/g of eWAT/mouse |
|---|---|---|
| ND | 0.29±0.25 x 105 | 0.62±0.45 x 105 |
| HFD-Veh | 1.43±0.086 x 106 | 0.77±0.015 x 106 |
| HFD-STF | 0.76±0.13 x 106 | 0.36±0.018 x 106 |
| HFD-Veh/ND: Fold (P value) | 49.63 (0.0041) | 12.54 (0.0045) |
| HFD-STF/HFD-Veh: Fold (P value) | 0.54 (0.053) | 0.46 (0.0033) |

**B**

**F4/80+CD11B+CD11C+**

|  | Percentage | Total cell#/eWAT/mouse | Cell#/g of eWAT/mouse |
|---|---|---|---|
| ND | 14.59±6.92 | 0.25±0.16 × 104 | 0.59±0.23 × 104 |
| HFD-Veh | 52.95±11.25 | 0.75±0.11 × 106 | 4.11±0.96 × 105 |
| HFD-STF | 37.65±12.65 | 0.27±0.05 × 106 | 1.32±0.38 × 105 |
| HFD-Veh/ND: Fold (P value) | 3.63 (0.007) | 299.58 (0.00036) | 70.06 (0.0018) |
| HFD-STF/HFD-Veh: Fold (P value) | 0.71 (0.19) | 0.36 (0.0027) | 0.32 (0.009) |

**C**

**F4/80+CD11B+CD11C+CD206+**

|  | Percentage | Total cell#/eWAT/mouse | Cell#/g of eWAT/mouse |
|---|---|---|---|
| ND | 13.46±7.15 | 0.21±0.13 × 104 | 0.63±0.27 × 104 |
| HFD-Veh | 50.45±8.25 | 7.13±0.74 × 105 | 4.19±0.05 × 105 |
| HFD-STF | 38.25±9.15 | 2.80±0.19 × 105 | 1.35±0.07 × 105 |
| HFD-Veh/ND: Fold (P value) | 3.75 (0.0042) | 337.56 (7.79E-05) | 75.0 (0.0008) |
| HFD-STF/HFD-Veh: Fold (P value) | 0.76 (0.16) | 0.39 (0.00062) | 0.35 (0.0049) |

**d**

**F4/80+CD11B+CD206+CD11C-**

|  | Percentage | Total cell#/eWAT/mouse | Cell#/g of eWAT/mouse |
|---|---|---|---|
| ND | 72.35±2.25 | 2.03±1.72 × 104 | 4.36±3.13 × 104 |
| HFD-Veh | 23.35±3.55 | 3.37±0.71 × 105 | 1.80±0.24 × 105 |
| HFD-STF | 28.95±5.25 | 2.14±0.017 × 105 | 1.02±0.13 × 105 |
| HFD-Veh/ND: Fold (P value) | 0.32 (3.55E-05) | 16.61 (0.0017) | 4.13 (0.0038) |
| HFD-STF/HFD-Veh: Fold (P value) | 1.24 (0.20) | 0.64 (0.40) | 0.57 (0.0076) |

immunofluorescent staining of CD9 and Trem2 was drastically increased in the eWAT under HFD, relative to normal chow (*Figure 6—figure supplement 1D-E*), but was profoundly diminished in STF-treated DIO mice (*Figure 6E–G*). Similarly, we observed a similar trend in the effect of STF on the immunofluorescent staining of CD9 and Trem2 in BAT (*Figure 6—figure supplement 1F*) and iWAT

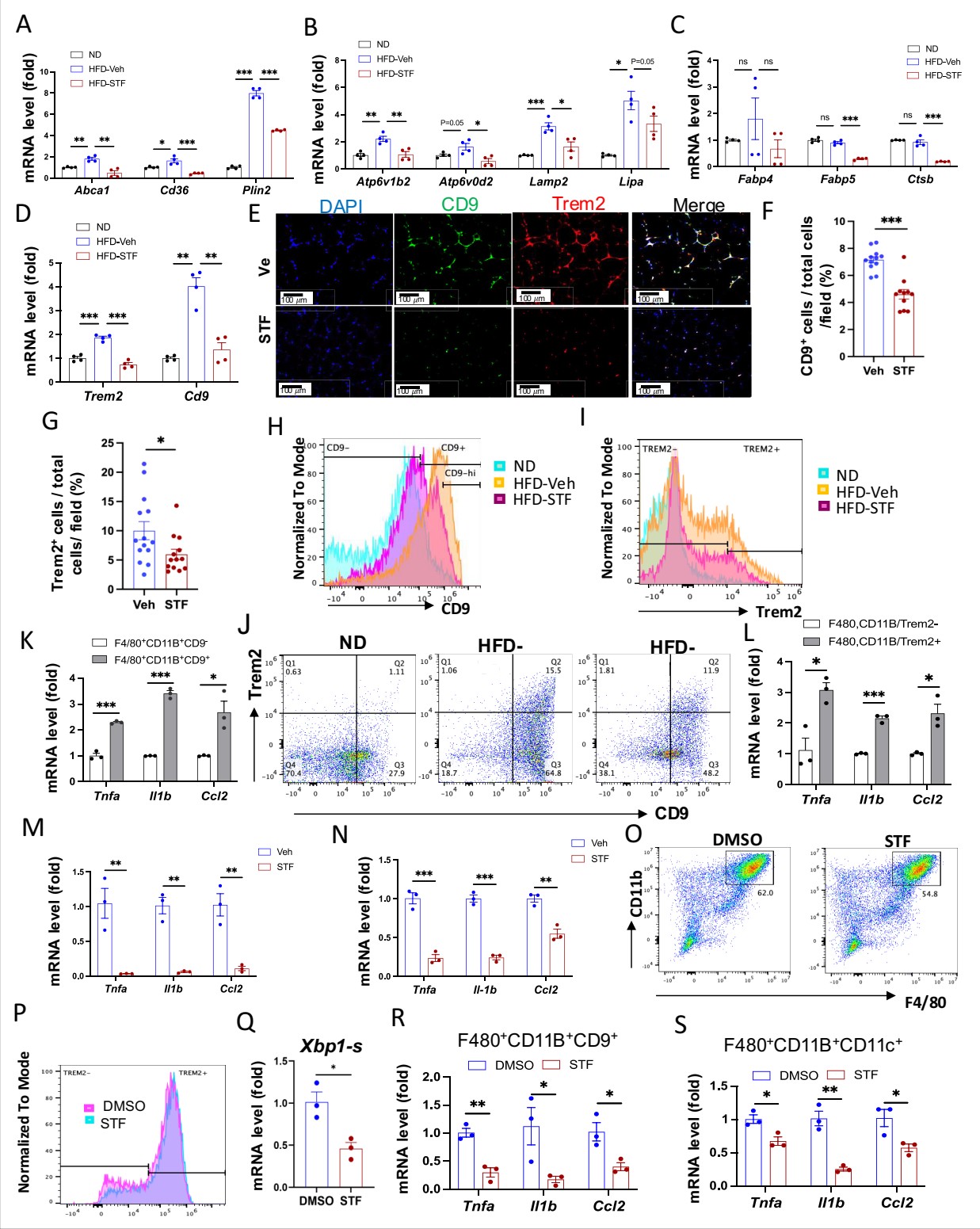

**Figure 6.** STF suppresses the accumulation of metabolically active CD9 or Trem2-expressing ATMs in obesity. (**A–D**) mRNA levels of indicated genes in the SVFs of eWAT from ND, HFD-veh and HFD-STF mice, assessed by qRT-PCR. The results are expressed as fold change and are representative of three independent experiments. (**E–G**) Immunofluorescent staining of CD9 and Trem2 in the eWAT sections of DIO mice treated with vehicle or STF. Representative images shown (**E**) and percentages of the numbers of CD9-positive cells (**F**) and Trem2-positive cells (**G**) over total cells/field. (**H**) Analysis of CD9-positive ATMs (F4/80+CD11b+CD9+) from eWAT SVFs of mice with an overlay of gated CD9 marker for ND (blue), HFD-Veh (yellow) and HFD-STF (pink) shown. Data in Y-axis was presented as 'Normalized to mode' for percentages. Complete gating path was shown in *Figure 5—*

*Figure 6 continued on next page*

*Figure 6 continued*

*figure supplement 2A*. (**I**) Analysis of Trem2-positive ATMs (F4/80⁺CD11b⁺Trem2⁺) from eWAT SVFs of mice with an overlay of gated Trem2 marker for ND (blue), HFD-Veh (yellow) and HFD-STF (pink) shown. Data in Y-axis was presented as 'Normalized to mode' for percentages. Complete gating path was shown in *Figure 5—figure supplement 2A*. (**J**) Flow cytometry analysis of CD9⁺ and Trem2⁺ ATMs from eWAT SVFs of ND, HFD-Veh and HFD-STF mice, gated against CD9 and Trem2 antibodies. Complete gating path was shown in *Figure 5—figure supplement 2A*. (**K–L**) mRNA levels of indicated proinflammatory genes in the indicated cells sorted from SVFs of eWAT of DIO mice, assessed by qRT-PCR: for CD9⁺ vs CD9⁻ ATMs (**K**) and for Trem2⁺ vs Trem2⁻ ATMs (**L**). The results are expressed as fold change and are representative of three independent experiments. M-N. mRNA levels of indicated proinflammatory cytokine/chemokine genes in the CD9⁺ ATMs (F4/80⁺CD11b⁺CD9⁺) (**M**) and the Trem2⁺ ATMs (F4/80⁺CD11b⁺Trem2⁺) (**N**) sorted from SVFs of eWAT of DIO mice treated with vehicle or STF, assessed by qRT-PCR. The results were expressed as fold change and are representative of three independent experiments. O-P. Flow cytometry analysis of ATMs from SVFs of the eWAT from 20-week HFD-fed mice. SVFs were cultured and treated with 0.01% DMSO control or STF (30 µM) for 20 hr followed by staining and flow analysis, gated against F4/80 and CD11b antibodies (**O**) and further gated against Trem2 antibody (**P**). Data in Y-axis in P was presented as 'Normalized to mode' for percentages. The gating path followed the one as *Figure 5—figure supplement 2A*. Q-S. mRNA levels of indicated genes in CD9⁺ ATMs (**Q and R**) and in CD11C⁺ ATMs (**S**) sorted from SVFs cultured and treated with DMSO or STF as in O-P, assessed by qRT-PCR. Data were expressed as mean ± SEM and analyzed using the unpaired two-tailed Student's t-test between two samples or ANOVA with multiple comparisons. ∗p<0.05, ∗∗p<0.01, and ∗∗∗p<0.001.

The online version of this article includes the following figure supplement(s) for figure 6:

**Figure supplement 1.** STF suppresses metabolically active ATMs in DIO mice.

**Figure supplement 2.** STF suppresses CD9 + and Trem2 + ATMs in DIO mice.

---

(*Figure 6—figure supplement 1G-H*) slides from DIO mice treated with STF or Veh, although without reaching statistical significance. We then used flowcytometry to characterize the effect of STF on CD9- or Trem2-expressing ATM subpopulations. We initially examined the HFD-induced changes in these populations. As expected, the percentage (23.25 ± 8.75% (ND, n=8) vs 72.70 ± 2.60% (HFD, n=8), p=0.032, *Figure 6H*, *Table 2a*), total number (0.45±0.32 x 10⁴/eWAT/mouse (ND) vs. 1.04 ± 0.026 x 10⁶˙ /eWAT/mouse (HFD), 228-fold, p=0.00062, *Table 2a*), and density (1.04±0.51 x 10⁴ /g of fat pad (ND) vs. 0.56±0.032 x 10⁶ /g of fat pad (HFD), 54-fold, p=0.0034, *Table 2a*) of the CD9⁺ ATMs (F4/80⁺CD11b⁺CD9⁺) in the eWAT were increased under HFD. It was previously reported that CD63 is enriched in CD9⁺ ATMs (*Jaitin et al., 2019*); indeed, we found that up to 98% of CD9⁺ ATMs also expressed CD63 in HFD-fed mice (*Figure 6—figure supplement 2*). As expected, the percentage, total number, and density of CD9⁺CD63⁺ ATMs increased significantly in the eWAT (*Figure 6—figure supplement 2B* and *Supplementary file 2*; ND: n=8; HFD: n=8). Similarly, Trem2⁺ ATM subpopulation expanded markedly in the percentage, total number/eWAT/mouse, and density (number/g of fat pad/ mouse) under HFD (*Figure 6I*, *Table 2b*). In addition, we found that while only approximately 47% of the CD9⁺ ATMs (F4/80⁺CD11b⁺CD9⁺) express Trem2, an overwhelming majority (~86.7%) of Trem2⁺ ATMs (F4/80⁺CD11b⁺Trem2⁺) expresses CD9 (*Figure 6—figure supplement 2C*), corroborating a previous finding that Trem2⁺ expression emerges after CD9⁺ ATMs over the course of obesity (*Jaitin et al., 2019*). As expected, double-positive CD9⁺Trem2⁺ ATMs also expanded tremendously under HFD (*Figure 6J*, *Table 2c*). In DIO mice treated with STF, we observed a decrease in the percentages of CD9⁺, Trem2⁺, and CD9⁺Trem2⁺ ATMs relative to that of vehicle-treated mice (*Figure 6H–J*, *Table 2a–c*). Further, STF treatment significantly diminished the total number and density of CD9⁺, Trem2⁺, and CD9⁺Trem2⁺ ATM subpopulations all by over 60% (*Table 2a–c*).

The CD9-expressing ATM subpopulation under HFD was proposed to be pro-inflammatory (*Hill et al., 2018*; *Jaitin et al., 2019*). Indeed, we observed that compared to CD9⁻ ATMs sorted from the eWAT of DIO mice, the sorted CD9⁺ ATMs expressed significantly higher mRNA levels of pro-inflammatory genes *Tnfa*, *Il1b*, and *Ccl2* (*Figure 6K*). Similarly, the mRNA levels of *Tnfa*, *Il1b*, and *Ccl2* were also heightened in sorted Trem2⁺ ATMs relative to Trem2⁻ ATMs (*Figure 6L*). Notably, STF treatment of DIO mice markedly reduced the expression levels of *Tnfa*, *Il1b*, and *Ccl2* in CD9⁺ ATMs sorted from the eWAT (*Figure 6M*). A similar trend was obtained in sorted Trem2⁺ ATMs from the obese eWAT (*Figure 6N*). These results together demonstrated that STF treatment suppresses the subpopulations of metabolically active CD9 and Trem2-positive ATMs and adipose inflammation.

To investigate whether IRE1α inhibition acts on ATMs directly, we examined the effect of STF on the macrophage population and inflammation in SVF cells (freshly isolated from obese eWATs) treated with STF. We observed that STF treatment for 20 hours slightly but statistically significantly diminished the percentage of ATMs (F4/80⁺CD11b⁺; DMSO 56.03% vs. STF 49.30%, p=0.0024) as analyzed by flow cytometry (*Figure 6O*). When further gated against Trem2, we observed a trend similar to that

**Table 2.** Characterization of and the effect of STF treatment on CD9$^+$ and Trem2$^+$ ATMs in DIO mice.

The percentage, total cell number, and density of subpopulations of CD9$^+$ ATMs (A), Trem2$^+$ ATMs (B), and CD9$^+$ Trem2$^+$ ATMs(C) in eWAT from mice with ND, HFD-Veh, or HFD-STF. Data in a-c were obtained from two batches of four mice each and were expressed as mean ± SEM and analyzed using the unpaired two-tailed Student's t-test between two samples or ANOVA with multiple comparisons. ∗p<0.05, ∗∗p<0.01, and ∗∗∗p<0.001.

**A**

F4−80+CD11B+CD9+

|  | Percentage | Total cell#/eWAT/mouse | Cell#/g of eWAT/mouse |
|---|---|---|---|
| ND | 23.25±8.75 | 0.45±0.32 × 104 | 1.04±0.51 × 104 |
| HFD-Veh | 72.70±2.60 | 1.04±0.026 × 106 | 0.56±0.032 × 106 |
| HFD-STF | 55.50±13.20 | 0.41±0.027 × 106 | 0.20±0.037 × 106 |
| HFD-Veh/ND: Fold (P value) | 3.13 (0.032) | 228.14 (0.00062) | 54.08 (0.0034) |
| HFD-STF/HFD-Veh: Fold (P value) | 0.77 (0.16) | 0.39 (0.0035) | 0.35 (0.017) |

**B**

F4−80+CD11B+Trem2+

|  | Percentage | Total cell#/eWAT/mouse | Cell#/g of eWAT/mouse |
|---|---|---|---|
| ND | 13.33±11.58 | 0.98±0.046 × 103 | 2.99±1.12 × 103 |
| HFD-Veh | 21.60±5.00 | 3.04±0.52 × 105 | 1.68±0.42 × 105 |
| HFD-STF | 16.70±2.80 | 1.24±0.06 × 105 | 0.59±0.07 × 105 |
| HFD-Veh/ND: Fold (P value) | 1.62 (0.32) | 309.97 (0.00056) | 56.13 (0.0024) |
| HFD-STF/HFD-Veh: Fold (P value) | 0.77 (0.21) | 0.41 (0.0040) | 0.35 (0.011) |

**C**

F4−80+CD11B+CD9+Trem2+

|  | Percentage | Total cell#/eWAT/mouse | Cell#/g of eWAT/mouse |
|---|---|---|---|
| ND | 7.56±6.45 | 0.59±0.0090 × 103 | 1.75±0.56 × 103 |
| HFD-Veh | 19.95±4.45 | 2.81±0.46 × 105 | 1.55±0.37 × 105 |
| HFD-STF | 14.45±2.55 | 1.07±0.0041 × 105 | 0.51±0.065 × 105 |
| HFD-Veh/ND: Fold (P value) | 2.64 (0.052) | 479.54 (0.00047) | 88.59 (0.0021) |
| HFD-STF/HFD-Veh: Fold (P value) | 0.72 (0.14) | 0.38 (0.0029) | 0.33 (0.0092) |

for the overall ATMs in that STF treatment moderately but significantly decreased the percentage of Trem2$^+$ ATMs (F4/80$^+$CD11b$^+$Trem2$^+$; DMSO 68.67% vs. STF 61.93%, p=0.0013; *Figure 6P*). We also observed that STF mildly lessened the percentages of CD9$^+$ ATMs (F4/80$^+$CD11b$^+$CD9$^+$; DMSO 76.73% vs. STF 74.80%, p=0.013; *Figure 7—figure supplement 1A*) and of CD11C$^+$ ATMs (F4/80$^+$CD-11b$^+$CD11C$^+$; DMSO 81.23% vs. STF 77.63%, p=0.0040) (*Figure 7—figure supplement 1B*). We next investigated the effect of STF on the expression levels of pro-inflammatory genes in ATMs sorted from STF- or DMSO-treated SVF cells. As expected, STF treatment decreased spliced *Xbp1* mRNA levels (*Figure 6Q*, *Figure 7—figure supplement 1C, G, I*), reflecting the inhibition of IRE1α RNase activity, and significantly down-regulated the expression levels of pro-inflammatory genes (*Tnfa*, *Il1b*, and *Ccl2*) in CD9$^+$ ATMs (F4/80$^+$CD11b$^+$CD9$^+$; *Figure 6R*) and in CD11C$^+$ ATMs (F4/80$^+$CD11b$^+$CD11C$^+$; *Figure 6—figure supplement 1S*), relative to DMSO control-treated counterparts. Similarly, in CD9$^-$ ATMs (F4/80$^+$CD11b$^+$CD9$^-$) or CD11C$^-$ ATMs (F4/80$^+$CD11b$^+$CD11C$^-$) sorted from STF- or DMSO-treated SVFs, which displayed lower mRNA expression levels of *Tnfa*, *Il1b*, and *Ccl2* compared to their respective CD9$^+$ or CD11C$^+$ counterparts (*Figure 7—figure supplement 1D-E*), STF also down-regulated the mRNA levels of *Tnfa*, *Il1b*, and *Ccl2* (*Figure 7—figure supplement 1F, H*). Together, these results indicate that STF likely acts directly on ATMs to regulate inflammation.

## The CD11c⁺ ATMs are overlapping with but yet distinctly different from metabolically active ATMs in obesity

One important question on the obesity-associated ATM accumulation is the obscure relationship between the newly-identified CD9⁺ and Trem2⁺ ATM populations and the originally identified 'M1-like' CD11c⁺ ATMs (*Hill et al., 2018*; *Wu and Ballantyne, 2020*). To investigate this relationship, we analyzed these populations using flow cytometry (*Figure 7—figure supplement 2*). First, we asked whether the previously identified F4/80⁺CD11b⁺CD11c⁺ triple positive ATMs express signature markers of the newly identified ATM subpopulations (CD9 and/or Trem2) in DIO mice. Our results showed that approximately 80% (91.30 ± 0.40%) of CD11c⁺ ATMs (F4/80⁺CD11b⁺CD11c⁺) express CD9 (*Figure 7A* and *Table 3a*; ND: n=8; HFD: n=8). On the other hand, about 70. 4% of CD11c-negative ATMs (F4/80⁺CD11b⁺CD11c⁻) express CD9 (*Figure 7A*, *Table 3a*). We also found that 23.95% of F4/80⁺CD11b⁺CD11c⁺ ATMs express CD9⁻ʰⁱ while only 3.185% of CD11c-negative ATMs are positive for CD9⁻ʰⁱ (*Figure 7A*). Similarly, the percentage of cells expressing Trem2 was higher in the CD11c⁺ ATMs than in CD11c⁻ ATMs (43.55 ± 8.65% for CD11c⁺ vs. 24.35 ± 4.05%, albeit not reaching statistical significance (p=0.074), *Figure 7B*, and *Table 3a*). We also observed a higher percentage of the CD11c⁺ ATMs (30.05 ± 5.55%) expressing both CD9 and Trem2 than that of CD11c⁻ ATMs (8.35 ± 1.86%), p=0.066, (*Figure 7C* and *Table 3a*). These results revealed that CD9 and Trem2 are highly enriched but not exclusively expressed in the CD11c⁺ ATMs under HFD.

In reciprocal studies, we analyzed the distribution of CD11c expression in CD9⁺ and/or Trem2⁺ ATMs under HFD. We observed heightened percentages of CD9⁺, Trem2⁺, and CD9⁺Trem2⁺ ATMs that co-express CD11c (63.25 ± 10.85%, 78.25 ± 12.35%, and 78.65 ± 13.05%, respectively) relative to the CD9⁻, Trem2⁻, and CD9⁻Trem2⁻ ATMs that co-express CD11c (29.90 ± 9.10%, 49.55 ± 9.85%, or 24.60 ± 5.90%, respectively), in the eWAT of HFD-fed mice (*Figure 7D–F* and *Table 3b*; ND: n=8; HFD: n=8). We also observed that higher percentage of CD9ʰⁱ ATM subpopulation expresses CD11c than CD9⁺ ATMs (90.60 ± 1.4% vs 63.25 ± 10.25%, *Figure 7D*). These results showed that while CD11c expression is not exclusively expressed in CD9⁺, Trem2⁺, or CD9⁺Trem2⁺ ATMs, it is highly enriched in all of these subpopulations. Together, these results reveal that in the eWAT under HFD, the majority of CD11c⁺ or CD9⁺ ATMs expresses CD9 or CD11c, respectively. Indeed, up to half (45.45 ± 5.05%) of ATMs in the obese eWAT co-express CD9 and CD11c (F4/80⁺CD11b⁺CD9⁺CD11c⁺) (*Figure 7G* and *Table 3c*; ND: n=8; HFD: n=8). Notably, similar to its effects on CD9⁺ or CD11c⁺ ATMs (*Figure 6H–I*), STF treatment substantially decreased the percentage, total number /eWAT/mouse, and density (per gram of fat pad/mouse) of the CD9⁺CD11c⁺ ATMs in the eWAT of DIO mice (*Figure 7G*, *Table 3c*). As CD9⁺ and CD11c⁺ ATMs were reported to be pro-inflammatory, we next compared the expression levels of proinflammatory genes in these ATM subpopulations from HFD-fed mice and observed that the mRNA levels of *Tnfa*, *Il1b*, and *Ccl2* were all up-regulated in sorted CD9⁺CD11c⁻, CD9⁻CD11c⁺, and CD9⁺CD11c⁺ ATMs compared to the CD9⁻CD11c⁻ ATMs, with the CD9⁺CD11c⁺ ATMs expressing higher level of *Ccl2* and the CD9⁺CD11c⁻ ATMs expressing lower level of *Il1b* (*Figure 7H*), suggesting that these ATM subpopulations are all proinflammatory but yet heterogeneous in nature.

## Discussion

In this study, we have shown that the RNase activity of ER stress sensor IRE1α is up-regulated in the adipose tissue of mice with diet-induced obesity and that the increased IRE1α activity precedes the onset of diabetes in the mice with obesity. By pharmacologically inhibiting IRE1α using the chemical STF, we found that the inhibition of IRE1α activity is able to reverse HFD-induced insulin resistance and comorbidities, indicating that the increased IRE1α activity is an important contributor to obesity-associated insulin resistance. We showed that STF promotes thermogenesis and energy expenditure and counters diet-induced obesity. Our results also showed that the IRE1α inhibition by STF suppresses the accumulation of proinflammatory ATMs, including previously identified CD11c⁺ ATMs and the recently identified CD9⁺ ATMs, and adipose inflammation. Our study also reveals that the CD11c⁺ ATMs are largely overlapping with but yet non-identical to CD9⁺ ATMs in the eWAT under HFD.

ER stress and its sensor IRE1α are activated in adipose tissues of subjects with obesity. IRE1α activation is known to be linked to inflammation via its dual enzymatic activity and its control of transcriptional regulation (*Martinon et al., 2010*; *Lerner et al., 2012*; *Oslowski et al., 2012*) while

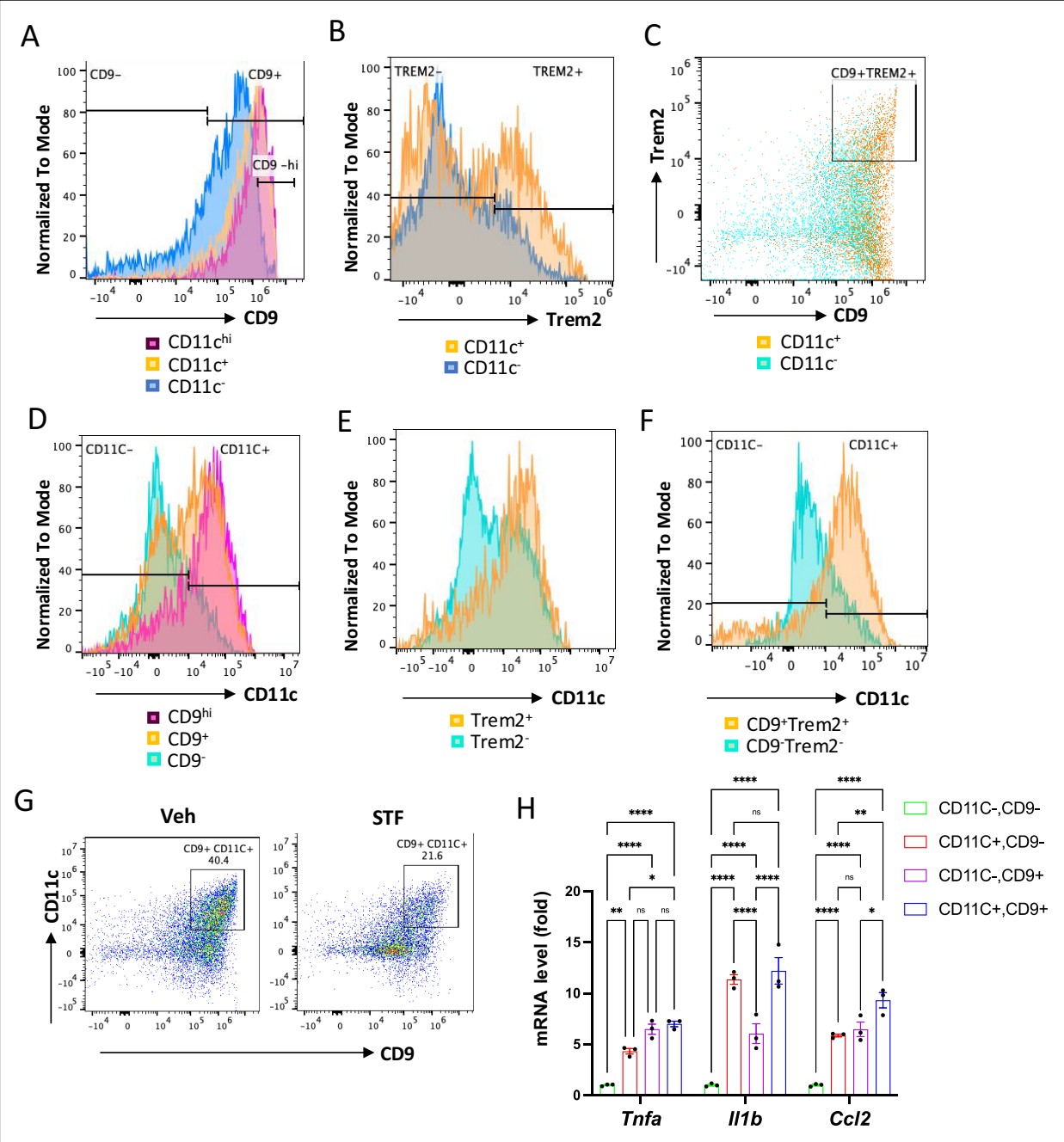

**Figure 7.** The relationship of CD11c⁺ ATMs with metabolically active CD9⁺ or Trem2⁺ ATMs in DIO mice. (**A**) Overlay of the flow cytometric analysis of percentages of CD9-positive signals among CD11C-ʰⁱ (pink), CD11C⁺ (yellow), and CD11C⁻ (blue) ATMs from eWAT SVFs of DIO mice. Data in Y-axis was presented as 'Normalized to mode' for percentages. (**B–C**) Overlay of the flow cytometric analysis of percentages of CD9-positive (**B**) and CD9- and Trem2-doubly positive (**C**) signals among CD11C⁺ (yellow) and CD11C⁻ (blue/green) ATMs from eWAT SVFs of DIO mice. (**D–F**) Overlay of the flow cytometric analysis of percentages of CD11-positive signals among CD9ʰⁱ, ⁺, ᵃⁿᵈ ⁻ ATMs (**D**), Trem2⁺ ᵃⁿᵈ ⁻ ATMs (**E**), and CD9/Trem2⁺ ᵃⁿᵈ ⁻ ATMs (**F**) from eWAT SVFs of DIO mice. Data in Y-axis was presented as 'Normalized to mode' for percentages. The gating strategies for A-F were shown in *Figure 7—figure supplement 2*. (**G**) Flow cytometry of CD9⁺ and CD11C⁺ ATMs from eWAT SVFs of DIO mice treated with vehicle or STF, gated against CD9 and CD11C antibodies. Complete gating path was shown in *Figure 5—figure supplement 2A*. (**H**) mRNA levels of indicated proinflammatory cytokine/chemokine genes in the indicated ATM subpopulations sorted from SVFs of eWAT of DIO, assessed by qRT-PCR. The results were expressed as fold change and were representative of three independent experiments. Data were expressed as mean ± SEM and analyzed using the unpaired two-tailed Student's t-test between two samples or ANOVA with multiple comparisons. *p<0.05, **p<0.01, and ***p<0.001.

The online version of this article includes the following figure supplement(s) for figure 7:

**Figure supplement 1.** STF suppresses pro-inflammatory cytokines genes expression in ATM in vitro.

*Figure 7 continued on next page*

*Figure 7 continued*

**Figure supplement 2.** Flow cytometry analysis strategy related to *Figure 7*.

macrophage activation and inflammation in adipose tissue is tightly associated with metabolic dysfunction. Targeting ER stress/IRE1α has thus been proposed for its potential in countering obesity and obesity-related metabolic disorders including insulin resistance and type 2 diabetes (*Hetz et al., 2013*; *Marciniak et al., 2022*). Indeed, recent studies using tissue-specific deletion of IRE1α in mouse have shown that embryonic IRE1α deletion in macrophage (*Shan et al., 2017*) and in adipocyte (*Chen et al., 2022*) prevents diet-induced insulin resistance and obesity. In the present study, we have showed that STF inhibition of IRE1α is capable of reversing existing insulin resistance and obesity and therefore validate the notion of targeting IRE1α inhibition as a potential therapeutic target for the treatment of obesity and related metabolic disorders.

In obese adipose tissue, IRE1α was activated in both adipocytes and ATMs. Activated IRE1α was reported to cause its RNase-mediated degradation of *Ppargc1a* mRNA and in turn suppression of PGC1α-dependent UCP1 level in adipocytes (*Chen et al., 2022*). Consistent with this finding, the present study observed that STF treatment decreases the level of IRE1α target RNA *Xbp1s* (*Figure 1G* and *Figure 1—figure supplement 1D*) and elevates the RNA levels of *Ppargc1a* and *Ucp1* in adipose tissue of DIO mice (*Figure 4L and N*). In the adipose tissue macrophage, IRE1α deletion in myeloid lineage using Ern1[fl/fl];Lyz2-cre was reported to induce the beiging of WAT and protect against diet-induced obesity by augmenting M2 macrophage population through its synthesis and secretion of catecholamine (*Shan et al., 2017*). It is well established that catecholamines play critical roles on thermogenesis through the activation of β-adrenoceptor signaling in the adipocyte tissue, which is critical for BAT activation and WAT remodeling. Catecholamines are normally synthesized and secreted from sympathetic nerves to induce adaptive thermogenesis in response to cold stimulation. It was reported

**Table 3.** Flow cytometry analysis of relationship between 'M1-like' ATMs and CD9[+] /Trem2[+] ATMs.
(A) The percentages of cells expressing CD9, Trem2 or both CD9 and Trem2 markers in CD11c[+] or CD11c[-] ATMs from eWATs of DIO mice. (B) The percentages of cells expressing CD11C marker in CD9[+], Trem2[+], or CD9[+] Trem2[+] ATMs versus their negative counterparts from eWATs of DIO mice. (C) The percentage, total cell number, and density of CD9[+] CD11c[+] ATMs from eWATs of DIO mice treated with vehicle or STF. Data in a-c were obtained from two batches of four mice each and were expressed as mean ± SEM and analyzed using the unpaired two-tailed Student's t-test between two samples or ANOVA with multiple comparisons. *p<0.05, **p<0.01, and ***p<0.001.

**A**

| Percentage of cells expressing | CD9 | Trem2 | CD9, Trem2 |
|---|---|---|---|
| F4/80+CD11b+CD11c+ | 91.30±0.40 | 43.55±8.65 | 30.05±5.55 |
| F4/80+CD11b+CD11c- | 70.40±1.0 | 24.35±4.05 | 8.35±1.86 |
| P value | 0.0027 | 0.074 | 0.066 |

**B**

| | CD9+ | CD9- | Trem2+ | Trem2- | CD9 +Trem2+ | CD9-Trem2- |
|---|---|---|---|---|---|---|
| Percentage of CD11c+ | 63.25±10.85 | 29.90±9.1 | 78.25±12.35 | 49.55±9.85 | 78.65±13.05 | 24.60±5.90 |
| p value | 0.14 | | 0.21 | | 0.063 | |

**C**

F4−80+CD11B+CD9+CD11c+

| | Percentage | Total cell #/mouse | Cell #/fat weight (g) |
|---|---|---|---|
| ND | 8.16±3.24 | 0.16±0.11 × 104 | 0.36±0.17 × 104 |
| HFD-Veh | 45.45±5.05 | 6.45±0.33 × 105 | 3.52±0.46 × 105 |
| HFD-STF | 29.40±7.80 | 2.14±0.21 × 105 | 1.03±0.23 × 105 |
| HFD-Veh/ND: Fold (pvalue) | 5.57 (0.025) | 416.29 (0.0026) | 98.75 (0.017) |
| HFD-STF/HFD-Veh: Fold (p value) | 0.65 (0.23) | 0.33 (0.008) | 0.29 (0.040) |

that M2 macrophages also produce catecholamines to sustain adaptive thermogenesis (*Nguyen et al., 2011*). However, this finding was disputed by a recent study showing that M2 macrophages do not synthesize catecholamines (*Fischer et al., 2017*). Importantly, in contrast to the previous report (*Shan et al., 2017*), our present study showed that STF treatment does not promote M2 macrophage in the adipose tissue (*Figure 5G* and *Table 1d*), arguing against the possibility that STF inhibition of IRE1α promotes thermogenesis by increasing M2 ATMs. Instead, STF treatment diminishes the M2 ATM population along with the decline of 'M1-like' CD11c$^+$ ATMs (*Figure 5F* and *Table 1b*). Notably, STF treatment also significantly diminished the newly identified CD9$^+$, Trem2$^+$, and dual CD9$^+$CD11c$^+$ ATM subpopulations (*Figure 6H–J* and *Table 2a–c*). Like CD11c$^+$ ATMs, CD9$^+$, Trem2$^+$, and dual CD9$^+$CD11c$^+$ ATM express higher levels of pro-inflammatory cytokine genes (*Figures 5K–L , and 7I*), indicative of their pro-inflammatory nature (*Hill et al., 2018*; *Jaitin et al., 2019*). STF treatment accordingly diminishes the expression levels of pro-inflammatory genes in both bulky adipose tissue (*Figure 5H*) and sorted ATMs (*Figures 5J and 6M–N*). Our in vitro experiments showed that STF treatment directly suppressed the ATM population and the expression of pro-inflammatory genes in these primary ATMs (*Figure 6—figure supplement 1Q-S*, *Figure 7—figure supplement 1*). Overall, our study indicates that STF inhibition of IRE1α reverses diet-induced adipose tissue inflammation and proinflammatory ATM subpopulations including the newly identified metabolically active CD9 ATMs.

Macrophage-derived proinflammatory cytokines such as TNFα and IL1b have been reported to directly suppress the induction of *Ucp1* expression in adipocytes and to impair cold-induced thermogenesis in rodents (*Sakamoto et al., 2013*; *Goto et al., 2016*; *Sakamoto et al., 2016*). Moreover, TNFα was shown to down-regulate the expression of β-adrenoceptor in adipocytes (*Hadri et al., 1997*; *Valentine et al., 2022*). HFD-induced obesity and chronic adipose inflammation are associated with the reduced expression of thermogenic/mitochondrial oxidation genes and the repression of thermogenesis (*Choi et al., 2015*). In present study, we observed that STF treatment increases the expression of UCP1, β$_3$-adrenoceptor, and other thermogenic genes in obese adipose tissue (*Figure 4L, N and O* and *Figure 4—figure supplement 2D-F*). These results suggest that the suppression of ATM population/inflammation by STF inhibition of IRE1α is associated with the reversal of the obesity-suppressed expression of thermogenic/mitochondrial oxidation genes in adipose tissue to promote thermogenesis, in conjunction with STF's action on the PGC1α expression in adipocytes.

We noticed that STF treatment protects against diet-induced obesity (*Figure 2A–B*) and metabolic improvements (*Figures 1–3*). It exists a possibility that STF may cause body weight reduction, which results in secondary beneficial effects on glucose and insulin sensitivity, as well as on ATM accumulation. We argue that such a possibility is unlikely as STF was previously reported to have no effect on body weight in two other studies: one in mouse model of atherosclerosis (*Tufanli et al., 2017*), and the other in Akita mouse model of beta cell death/failure (*Herlea-Pana et al., 2021*). Second, in our study, we did not observe any apparent difference in food intake (*Figure 4—figure supplement 1B*) that could otherwise affect body weight. Third, STF directly suppresses ATM population and pro-inflammatory gene expression. Finally, STF-treated DIO mice expended more energy than their vehicle-treated counterparts even at lower body weight (*Figure 4A–F*). Although it is known that energy expenditure per unit body weight appears to be lower in subjects with obesity than in lean controls because obesity is associated with reduced thermogenesis, subjects with obesity still expend more energy than the lean controls at the organismal level (*Prentice et al., 1996*; *Trayhurn and Arch, 2020*). If there exists leanness-associated thermogenesis increase per unit weight in STF-treated DIO mice, it cannot fully account for the energy expenditure increase at the organismal level as observed in STF-treated DIO mice, arguing for the notion that STF treatment stimulates thermogenesis which in turn leads to body weight reduce in a positive feedback loop. We also note that as specific IRE1α RNase inhibitors, STF and other members of hydroxy-aryl-aldehyde class have been studied in mouse models of multiple diseases (*Tufanli et al., 2017*; *Lebeaupin et al., 2018*; *Zhan et al., 2019*; *Herlea-Pana et al., 2021*; *Marek-Iannucci et al., 2022*), with beneficial effects and non-toxicity in general. Further studies on pharmacokinetic and toxicological parameters will be needed before any potential clinical trial is initiated.

Our study also shed new insight into the macrophage remodeling under obesity. Macrophage accumulation and activation in adipose tissue is known to contribute to insulin resistance in obesity. It was initially reported that adipose tissue macrophages (ATMs) undergo an M2 to M1 polarization switch under obesity. Recent studies have however indicated that obese ATMs do not follow

the canonical M1 polarization (*Xu et al., 2013*; *Kratz et al., 2014*; *Hill et al., 2018*; *Jaitin et al., 2019*). In the present study, we found that first, 'M1-like' ATMs (F4/80$^+$CD11b$^+$CD11c$^+$CD206$^-$ cells) significantly increase under HFD (*Figure 5F–G*, *Table 1b–c*); second, M2-like ATMs (F4/80$^+$CD11b$^+$ CD206$^+$CD11c$^-$ cells) also increase in total cell number and density in response to HFD (*Table 1d*), albeit to a smaller extent than the 'M1-like' ATMs, a phenomenon previously reported (*Xu et al., 2013*); and third, there is a significant increase (up to half of the ATMs gated with CD11c$^+$ and CD206$^+$) in ATMs co-expressing both CD11c$^+$ and CD206$^+$ in the adipose tissue of HFD-induced mice with obesity (*Figure 5G*, *Table 1c*), supporting a mixed phenotype rather than distinct M1 or M2 phenotype. Previous studies have identified such ATM subpopulation that express both M1 and M2 surface markers in both human and rodent adipose tissues under obesity (*Shaul et al., 2010*; *Wentworth et al., 2010*). These findings are consistent with the notion that ATMs do not follow the classic M2 to M1 switch. Instead, recent studies employing single-cell RNA-seq technology have revealed the expansion/emergence of ATMs that express cell surface markers CD9 or Trem2 (*Hill et al., 2018*; *Jaitin et al., 2019*). In the present study, we found that the percentage, density, and total number of CD9$^+$ ATMs (F4/80$^+$CD11b$^+$CD9$^+$) in the eWAT are drastically increased under HFD (*Figure 6H*, *Table 2a*). Similarly, Trem2$^+$ ATMs (F4/80$^+$CD11b$^+$Trem2$^+$) expand significantly in the eWAT of mice with obesity (*Figure 6I*, *Table 2b*), most of which also express CD9.

Our present study further investigated the relationship between the newly-identified CD9$^+$ ATMs and the originally identified 'M1-like' CD11c$^+$ ATMs as it remains unclear whether they are related or different ATM populations (*Hill et al., 2018*; *Wu and Ballantyne, 2020*). Our results showed that CD9 and Trem2 are highly enriched but not exclusively expressed in the CD11c$^+$ ATMs under HFD (*Figure 7A–C*). Reciprocally, CD11c expression is highly enriched but not exclusively expressed in CD9$^+$, Trem2$^+$, or CD9$^+$Trem2$^+$ ATMs (*Figure 7D–F*). As such, in the obese eWAT, up to half (49.95 ± 0.65%) of ATMs co-express CD9 and CD11c (*Figure 7G*). These results demonstrate that CD9$^+$ and CD11c$^+$ populations are highly overlapping. We further found that CD9$^+$CD11c$^-$, CD9$^-$CD11c$^+$, and CD9$^+$CD11c$^+$ ATMs are all pro-inflammatory as the mRNA levels of pro-inflammatory genes *Tnfa*, *Il1b*, and *Ccl2* are up-regulated in all these subpopulations than CD9$^-$CD11c$^-$ ATMs (*Figure 7H*). However, the mRNA levels of individual *Tnfa*, *Il1b*, or *Ccl2* are differentially expressed among CD9$^+$CD11c$^-$, CD9$^-$CD11c$^+$, and CD9$^+$CD11c$^+$ ATMs (*Figure 7H*), suggesting functional heterogeneity even among proinflammatory ATMs during obesity.

In summary, our present studies reveal two significant findings. First, we confirmed that the newly identified populations of proinflammatory ATMs (i.e. CD9$^+$ and Trem2$^+$) are induced under HFD-mediated obesity and determined their relationships with the originally identified 'M1-like' CD11c$^+$ ATMs. Second, we showed that STF inhibition of IRE1α reverses the accumulation of pro-inflammatory ATM populations including the new metabolically active CD9$^+$ and Trem2$^+$ ATMs and adipose tissue inflammation to promote thermogenesis and insulin sensitivity. These studies validate the potential of IRE1α inhibitors for the prevention and treatment of obesity, insulin resistance and comorbidities, which sets the foundation for further clinical development of IRE1α inhibitors as a therapeutic direction for diabetes and obesity.

## Materials and methods

### Animal studies

C57BL/6 J mice were obtained from Jackson laboratory (Bar Harbor, ME). Mice were kept in an animal facility with a 12 hr light/dark cycle and temperature of 22 °C. All animals had access to diet and water ad libitum. All procedures involving animals were approved by the Institutional Animal Care and Use Committee of the University of Oklahoma Health Science Center. All experiments were performed with age-matched male mice. The C57BL/6 J male mice were fed a HFD (60% kcal fat, Bio-Serv, NJ, USA) at the age of 5–6 weeks. At the 8-week HFD feeding, mice were randomly divided into two groups and were administered with either vehicle (10% DMSO) or STF (10 mg/kg body weight in 10% DMSO) once daily through the route of intraperitoneal injection. Blood glucose levels were measured using the OneTouch Ultra2 glucometer after fasting for 6 hr. Body weights were measured weekly.

## Glucose tolerance test and insulin tolerance test

Intraperitoneal glucose tolerance test (ipGTT) was performed after 16 hr overnight fasting. Blood glucose levels were measured at 0, 15-, 30-, 60-, and 120 min glucometer (OneTouch Ultra 2 Meter) after an intraperitoneal administration of glucose at dose of 1.5 g/kg body weight. Intraperitoneal insulin tolerance test (ipITT) was performed after 6 hr fasting. Blood glucose levels were measured at 0, 15, 30, 60, and 120 min after an intraperitoneal administration of human insulin at dose of 1.2 IU/kg body weight.

## Biochemical analysis

Fasting serum insulin (ALPCO, NH, USA), serum cholesterol TG (Cayman Chem., MI, USA) and FFAs (Bioassay system, CA, USA) were determined by ELISA.

## Body composition assessment

Body lean and fat composition and total body weight were determined by EchoMRI test.

## Indirect calorimetry measurements

The mice were individually housed in chambers of a Promethion Core Monitoring system (Sable Systems, Las Vegas, NV, USA). After a 1-day acclimation period, their oxygen consumption and carbon dioxide production were measured for the next 3 consecutive days. The respiratory exchange ratio and energy expenditure were calculated using standard equations.

## Adipose stromal vascular fraction (SVF) isolation and flow cytometry

Mice were sacrificed and the epididymal visceral white adipose tissue (eWAT) was immediately washed by PBS. The tissue was placed in 10 cm petri dish and excised into tiny bits with scissors at room temperature. Transfer the fine chopped tissue pieces in 10 mL plain DMEM containing Collagenase P (0.45 mg/mL) in 50 mL tubes. Keep the tubes in 37 °C incubator shaker at a speed of 200 rpm for 30–40 min. Add 5 mL of media containing FBS to each tube and pipette well and spin at 700 × $g$ rcf for 10 min at 4 °C. Remove gently all the supernatants, and the remaining pellet is stromal vascular fraction (reddish brown –containing RBC).

For flow cytometry, cells were resuspended in sterile FACS buffer, then filter through 70µm strainer and centrifuge at 1000 rpm at 4 °C for 5 minutes. Resuspend the pellet in 2 mL of 1 X RBC lysis buffer solution (Sigma) and incubated on ice for 2–3 min. Add 10 mL of PBS and spin the cells at 1100 rpm for 5 min. The resulting cell suspension was incubated with Fc Block (BD Biosciences) prior to staining with conjugated antibodies for 20 min at 4 °C followed by two washes in FACS buffer. Samples were stained using the following antibodies: PerCP-conjugated anti-mouse CD45 (Cat# 110725, RRID:AB_893347), FITC-conjugated anti-mouse F4/80 (Cat# 123108, RRID:AB_893502), PE-conjugated anti-mouse/human CD11b (Cat# 101207, RRID:AB_312790), BV421-conjugated anti-mouse CD11c, (Cat# 117330, RRID:AB_11219593), Alexa Fluor 700-conjugated anti-mouse CD206 (Cat# 141734, RRID:AB_2629637), PE Dazzle594-conjugated anti-mouse CD9 (Cat# 124821, RRID:AB_2800601), and PE-Cy7-conjugated anti-mouse CD63 (Cat# 143909, RRID:AB_2565499), purchased from Biolegend. APC-conjugated Trem2 (Cat# FAB17291N, RRID:AB_3646995, R&D Systems). Fluorescence Minus One (FMO) controls were included in the flow experiments. Cells were analyzed using Aurora Spectral Flow Cytometer (Cytek) and FlowJo software (FlowJo 10.10, LLC). Sorting was performed using MoFloXDP (Beckman Coulter). Single cells were sorted into 1.5 mL tube containing 50 µL of DMEM medium. Immediately after sorting, tubes were spun down and removed the medium, snap-frozen on dry ice and stored at −80 °C until further processing.

## SVF culture and treatment in vitro

Freshly isolated SVF portions were cultured in RPMI 1640, supplemented with 10% FBS and 1% penicillin/streptomycin (PS; RPMI/FBS/PS). 2 hr after seeding for a density at 2x10$^6$ cells/10 cm-dish, SVF cells were treated with 0.01% DMSO control or 30µM STF for 20 hr. Cells were collected and labeled with the antibodies as above followed by FACS analysis or sorting.

## Histology and immunohistochemistry

Adipose tissue and liver were dissected and immediately fixed in 4% paraformaldehyde (Sigma–Aldrich), paraffin embedded, and stained with hematoxylin and eosin. For Immunofluorescence staining, the

eWAT and iWAT paraffin sections (the section thickness is 5–10 μm) were blocked with 1% BSA, 0.2% Triton PBS, then incubated with primary antibody F4/80 (Cat#:70076, RRID:AB_2799771, Cell Signaling Technology), CD9 (Cat#:ab223052, RRID:AB_2922392, Abcam), Trem2 (Cat#: MAB17291, RRID:AB_2208679, R&D Systems) UCP1(Cat#:ab10983, RRID:AB_2241462, Abcam) overnight at 4 °C after deparaffinization. The slides were washed thrice with PBS with 0.2% Triton and then incubated in Alexa Fluor 488-, 555-, and 647- conjugated secondary antibodies (Jackson ImmunoResearch, PA), DAPI (0.5 mg/mL) for 1.5 hr at room temperature. Tissue sections were imaged on the Olympus Fluoview 1000 laser-scanning confocal microscope (Center Valley, PA) and quantified with Image-J histogram software.

## RNA extraction and RT-PCR

Total RNA (2 μg) was isolated from tissues or cells using TRIzol reagents (Life Technologies) or total RNA (0.1–0.5 μg) from sorted cells using E.Z.N.A. total RNA kit (Omega Bio-Tek, Norcross, GA) isolated and reverse transcribed using oligo d(T) primers (New England Biosystems) and SuperScript IV reverse transcription kit (Applied Biosystems, Foster City, CA). qPCR was performed in a CFX96 Touch Real-Time PCR detection system using SYBR Green mix (Applied Biosystems, Foster City, CA). The amplification program was as follows: initial denaturation at 95 °C for 15 min, followed by 40 cycles of 95 °C for 15 s, 60 °C for 1 min, and 40 °C for 30 s. Relative mRNA expression was determined by the ΔΔCt method normalized to *Tbp* mRNA. The sequences of primers used in this study are found in *Supplementary file 3*. Regular RT-PCR was performed and the products were resolved by agarose gel electrophoresis. The unspliced and spliced *Xbp1* mRNA levels were quantified using ImageJ software (National Institutes of Health, Bethesda, MD).

## Western blotting

Proteins were extracted with RIPA buffer containing protease and phosphatase inhibitors (Thermo) and then centrifuged for 10 min at $10,000 \times g$. Total protein concentration in the cell lysate was determined by Pierce BCA Protein Assays. Samples of ~30 μg protein prepared with (reducing) 5% β-mercaptoethanol were separated on a 4–12% Bis-Tris NuPAGE gels (Invitrogen) and transferred to polyvinylidene-fluoride membrane for 1 hr at 4 °C at 40 V. The membrane was blocked in TBS (10 mm Tris-HCl, pH 7.4, 150 mm NaCl) containing 3% BSA for 1 hr at room temperature and was probed with primary antibodies followed by the appropriate HRP-conjugated secondary antibodies (1:5000; Santa Cruz Biotechnology, CA, USA). The primary antibodies were: Phospho-Akt (Ser473; 4060 S, RRID:AB_2315049, Cell Signaling Technology), Akt (pan) (4685 S, RRID:AB_2225340, Cell Signaling Technology), GAPDH (sc-365062, RRID:AB_10847862, Santa Cruz Biotechnology).

## Statistical analysis

All data are presented as the mean ± SEM of the indicated number of replicates. Data were analyzed using the unpaired two-tailed Student's t-test or ANOVA with multiple comparisons (Prism 10.3.1) and $p<0.05$ was considered statistically significant. For energy expenditure experiments, the data was analyzed for group effect (ANOVA) along with the mass and interaction effects as necessary (generalized linear model, GLM; *Mina et al., 2018*).

## Acknowledgements

We thank the Diabetes CoBRE histology and Imaging (5P30GM122744) and Molecular Biology and Cytometry Research (P30CA225520) Cores at OUHSC for assistance on histology and microscopy use. This work was supported by grants from the NIH (R01DK116017, R01DK128848, and R01DK130174) to WW.

# Additional information

## Funding

| Funder | Grant reference number | Author |
|---|---|---|
| National Institutes of Health | R01DK116017 | Weidong Wang |
| National Institutes of Health | R01DK128848 | Weidong Wang |
| National Institutes of Health | R01DK130174 | Weidong Wang |

The funders had no role in study design, data collection and interpretation, or the decision to submit the work for publication.

## Author contributions

Dan Wu, Data curation, Formal analysis, Validation, Investigation, Visualization, Methodology, Writing – original draft, Project administration, Writing – review and editing; Venkateswararao Eeda, Formal analysis, Methodology; Zahra Maria, Data curation, Investigation, Methodology; Komal Rawal, Oana Herlea-Pana, Ram Babu Undi, Investigation, Methodology; Audrey Wang, Writing – review and editing; Hui-Ying Lim, Supervision, Writing – original draft, Writing – review and editing; Weidong Wang, Conceptualization, Resources, Data curation, Formal analysis, Supervision, Funding acquisition, Validation, Investigation, Writing – original draft, Project administration, Writing – review and editing

## Author ORCIDs

Dan Wu (ID) http://orcid.org/0009-0002-8389-6107
Venkateswararao Eeda (ID) https://orcid.org/0000-0003-2740-047X
Zahra Maria (ID) http://orcid.org/0000-0002-7279-3279
Komal Rawal (ID) https://orcid.org/0000-0001-6707-3392
Audrey Wang (ID) https://orcid.org/0009-0005-4736-5634
Oana Herlea-Pana (ID) https://orcid.org/0000-0002-3235-6795
Ram Babu Undi (ID) https://orcid.org/0000-0001-5061-7066
Hui-Ying Lim (ID) https://orcid.org/0000-0001-8084-6337
Weidong Wang (ID) https://orcid.org/0000-0003-3619-0953

Reviewer #1 (Public review): https://doi.org/10.7554/eLife.100581.4.sa1
Reviewer #3 (Public review): https://doi.org/10.7554/eLife.100581.4.sa2
Author response https://doi.org/10.7554/eLife.100581.4.sa3

# Additional files

## Supplementary files

Supplementary file 1. Effect of STF treatment on "M1-like" and "M2-like" ATMs in obesity. The total cell number and density of populations from $CD11C^+CD206^-$ ATMs in the eWATs of mice with ND, HFD-Veh, or HFD-STF (a), $CD11C^-CD206^-$ ATMs in the eWATs of mice with ND, HFD-Veh, or HFD-STF (b), and $F4/80^-CD11B^-$ ATMs in the eWATs of DIO mice treated with Veh or STF (c). Data in a-c were obtained from 2 batches of 4 mice each and are the mean ± SEM. $*P<0.05$, $**P<0.01$, and $***P<0.001$.

Supplementary file 2. Characterization of and the effect of STF treatment on $CD9^+$ ATMs in DIO mice. The percentage, total cell number, and density of populations from $CD9^+CD63^+$ ATMs in the eWATs of mice with ND, HFD-Veh, or HFD-STF. Data were obtained from 2 batches of 4 mice each and are the mean ± SEM. $*P<0.05$, $**P<0.01$, and $***P<0.001$.

Supplementary file 3. Primers used in this work.

MDAR checklist

### Data availability

All data generated or analysed during this study are included in the manuscript and supporting files.

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
