## [Editor Report · eLife Assessment]

The study presents **important** findings on inositol-requiring enzyme (IRE1α) inhibition on diet-induced obesity (overnutrition) and insulin resistance where IRE1α inhibition enhances thermogenesis and reduces the metabolically active and M1-like macrophages in adipose tissue. The evidence supporting the conclusions is **convincing**. The work will be of interest to cell biologists and biochemists working in metabolism, insulin resistance and inflammation with a broad eLife readership.

---

## [Referee Report · Reviewer #1 (Public review)]

First, the authors confirm the up-regulation of the main genes involved in the three branches of the Unfolded Protein Response (UPR) system in diet-induced obese mice in AT, observations that have been extensively reported before. Not surprisingly, IRE1a inhibition with STF led to an amelioration of the obesity and insulin resistance of the animals. Moreover, non-alcoholic fatty liver disease was also improved by the treatment. More novel are their results in terms of thermogenesis and energy expenditure, where IRE1a seems to act via activation of brown AT. Finally, mice treated with STF exhibited significantly fewer metabolically active and M1-like macrophages in the AT compared to those under vehicle conditions. Overall, the authors conclude that targeting IRE1a has therapeutical potential for treating obesity and insulin resistance.

The study has some strengths, such as the detailed characterization of the effect of STF in different fat depots and a thorough analysis of macrophage populations. However, the lack of novelty in the findings somewhat limits the study´s impact on the field.

---

## [Referee Report · Reviewer #3 (Public review)]

Summary:

The manuscript by Wu D. et al. explores an innovative approach in immunometabolism and obesity by investigating the potential of targeting macrophage Inositol-requiring enzyme 1α (IRE1α) in cases of overnutrition. Their findings suggest that pharmacological inhibition of IRE1α could influence key aspects such as adipose tissue inflammation, insulin resistance, and thermogenesis. Notable discoveries include the identification of High-Fat Diet (HFD)-induced CD9+ Trem2+ macrophages and the reversal of metabolically active macrophages' activity with IRE1α inhibition using STF. These insights could significantly impact future obesity treatments.

Strengths:

The study's key strengths lie in its identification of specific macrophage subsets and the demonstration that inhibiting IRE1α can reverse the activity of these macrophages. This provides a potential new avenue for developing obesity treatments and contributes valuable knowledge to the field.

Weaknesses:

The research lacks an in-depth exploration of the broader metabolic mechanisms involved in controlling diet-induced obesity (DIO). Addressing this gap would strengthen the understanding of how targeting IRE1α might fit into the larger metabolic landscape.

Impact and Utility:

The findings have the potential to advance the field of obesity treatment by offering a novel target for intervention. However, further research is needed to fully elucidate the metabolic pathways involved and to confirm the long-term efficacy and safety of this approach. The methods and data presented are useful, but additional context and exploration are required for broader application and understanding.

Comments on revisions:

The authors have satisfactorily addressed all of my previous concerns.

---

## [Author Response]

The following is the authors’ response to the previous reviews.

**Public Reviews:**

**Reviewer #1 (Public review):**
First, the authors confirm the up-regulation of the main genes involved in the three branches of the Unfolded Protein Response (UPR) system in diet-induced obese mice in AT, observations that have been extensively reported before. Not surprisingly, IRE1a inhibition with STF led to an amelioration of the obesity and insulin resistance of the animals. Moreover, non-alcoholic fatty liver disease was also improved by the treatment. More novel are their results in terms of thermogenesis and energy expenditure, where IRE1a seems to act via activation of brown AT. Finally, mice treated with STF exhibited significantly fewer metabolically active and M1-like macrophages in the AT compared to those under vehicle conditions. Overall, the authors conclude that targeting IRE1a has therapeutical potential for treating obesity and insulin resistance.The study has some strengths, such as the detailed characterization of the effect of STF in different fat depots and a thorough analysis of macrophage populations. However, the lack of novelty in the findings somewhat limits the study´s impact on the field.

We thank the reviewer for the appreciation of our findings. We would use the opportunity to highlight several novelties. First, we characterized the relationship between the newly discovered CD9^+^ ATMs and the “M1-like” CD11c+ ATMs. Second, we demonstrated that M2 macrophage population was not reduced but instead increased in adipose tissue in obesity. Third, IRE1 inhibition does not improve thermogenesis by boosting M2 population, but instead, IRE1 inhibition suppresses pro-inflammatory macrophage populations including the M1-like ATMs.

**Reviewer #3 (Public review):**
Summary:The manuscript by Wu D. et al. explores an innovative approach in immunometabolism and obesity by investigating the potential of targeting macrophage Inositol-requiring enzyme 1α (IRE1α) in cases of overnutrition. Their findings suggest that pharmacological inhibition of IRE1α could influence key aspects such as adipose tissue inflammation, insulin resistance, and thermogenesis. Notable discoveries include the identification of High-Fat Diet (HFD)-induced CD9^+^ Trem2+ macrophages and the reversal of metabolically active macrophages' activity with IRE1α inhibition using STF. These insights could significantly impact future obesity treatments.Strengths:The study's key strengths lie in its identification of specific macrophage subsets and the demonstration that inhibiting IRE1α can reverse the activity of these macrophages. This provides a potential new avenue for developing obesity treatments and contributes valuable knowledge to the field.Weaknesses:The research lacks an in-depth exploration of the broader metabolic mechanisms involved in controlling diet-induced obesity (DIO). Addressing this gap would strengthen the understanding of how targeting IRE1α might fit into the larger metabolic landscape.

We thank the reviewer for the appreciation of strengths in our manuscript. In particular, we appreciate the reviewer’s recommendation on the exploration of broader metabolic landscape, such as the effect of IRE1 inhibition on non-adipose tissue macrophages and metabolism. We agree that achieving these will certainly broaden the therapeutic potential of IRE1 inhibition to larger metabolic disorders and we will pursue these explorations in future studies.

Impact and Utility:The findings have the potential to advance the field of obesity treatment by offering a novel target for intervention. However, further research is needed to fully elucidate the metabolic pathways involved and to confirm the long-term efficacy and safety of this approach. The methods and data presented are useful, but additional context and exploration are required for broader application and understanding.Comments on revisions:The author has revised the manuscript and addressed the most relevant comments raised by the reviewers. The paper is now significantly improved, though two minor issues remain.(1) Studies were limited to male mice; this should be mentioned in the paper's Title.

Thanks for comment. We have modified the title to reflect the male mice only.

(2) Please include the sample size (n=) in all provided tables in the main manuscript and supplementary tables.

We have included the sample size in the main manuscript.